# LiteGS: A High-performance Framework to Train 3DGS in Subminutes via System and Algorithm Codesign

## Abstract

3D Gaussian Splatting (3DGS) has emerged as promising alternative in 3D representation. However, it still suffers from high training cost. This paper introduces LiteGS, a high performance framework that systematically optimizes the 3DGS training pipeline from multiple aspects. At the low-level computation layer, we design a "warp-based raster" associated with two hardware-aware optimizations to significantly reduce gradient reduction overhead. At the mid-level data management layer, we introduce dynamic spatial sorting based on Morton coding to enable a performant "Cluster-Cull-Compact" pipeline and improve data locality, therefore reducing cache misses. At the top-level algorithm layer, we establish a new robust densification criterion based on the variance of the opacity gradient, paired with a more stable opacity control mechanism, to achieve more precise parameter growth. Experimental results demonstrate that LiteGS accelerates the original 3DGS training by up to 13.4x with comparable or superior quality and surpasses the current SOTA in lightweight models by up to 1.5x speedup. For high-quality reconstruction tasks, LiteGS sets a new accuracy record and decreases the training time by an order of magnitude.

## 1 Introduction

As an emerging technique for scene representation and novel view synthesis, 3DGS has achieved remarkable breakthroughs in computer graphics and computer vision due to its high rendering quality and rapid rendering speed (Kerbl et al., 2023). This technique represents 3D scenes using millions of anisotropic 3D Gaussian primitives to obtain photorealistic rendering results, demonstrating immense potential in fields such as autonomous driving, virtual reality, and digital twins (Lin et al., 2024; Liu et al., 2024; Tu et al., 2025; Kung et al., 2025). Although 3DGS rendering is very fast, the training process could take tens of minutes or even hours, which limits its application. Existing efforts have attempted to reduce the number of Gausians or simplify certain computation to accelerate 3DGS training, but are still unable to mitigate the main performance bottlenecks (Mallick et al., 2024; Durvasula & Chai, 2024; Liao et al., 2025).

A deeper analysis reveals that these bottlenecks originate from multiple aspects. At the low-level GPU system layer, the gradient reduction process undergoes high data conflicts, causing severe GPU under-utilization. At the data management layer, millions of Gaussian primitives are stored in a disordered manner, leading to poor spatial locality, drastically increasing cache misses and reducing hardware efficiency. At the algorithm layer, its non-robust densification strategy brings suboptimal geometric growth. To systematically address these issues, this paper proposes a high performance open-sourced framework, *LiteGS*. We move beyond optimizations at a single level and instead conduct a thorough enhancement of the entire 3DGS training workflow, from low-level GPU computation and mid-level data management to high-level algorithm design.

At the GPU system layer, we propose a new "warp-based raster" paradigm based on the principle of "one warp per tile". This approach simplifies the intra-tile gradient aggregation into a single intra-warp reduction. To mitigate the increased register pressure caused by thread-to-multipixel mapping, we introduce a scanline algorithm and employ mixed precision. This design not only significantly

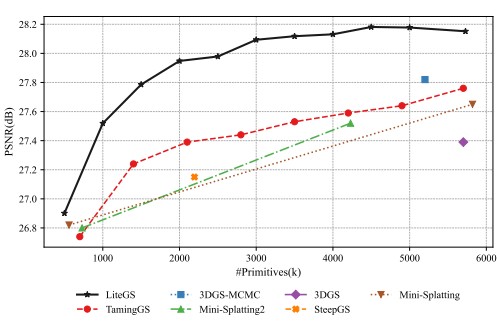 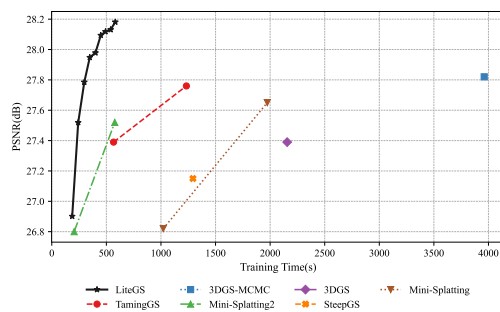

(a) PSNR versus primitive count. LiteGS achieves comparable or higher quality than state-of-the-art methods using only one-third the number of primitives.

(b) PSNR versus training time. LiteGS converges to high-quality results over 10x faster than prior works, demonstrating strong advantages in both compactness and training efficiency.

Figure 1: Reconstruction quality under varying model budgets on the Mip-NeRF 360 garden scene.

reduces the cost in gradient computation, but also allows the algorithm to efficiently perform pixel-level statistics gathering.

At the data management layer, we introduce a "Cluster-Cull-Compact" pipeline that utilizes Morton coding to perform dynamic spatial sorting for Gaussian primitives at an extremely low computational cost. After reordering, we can perform culling and memory compaction at the cluster granularity with better locality to notably reduce cache misses and warp divergence, making it an orthogonal optimization to the sparse gradient update strategies.

At the algorithm layer, our strategy discards the original ambiguous metric and instead adopts the more robust variance of per-pixel opacity gradient as the core criterion for densification. High variance can more accurately identify the underfit area. The cheap computation of this metric is a direct benefit of the efficient statistical capabilities provided by our underlying rasterizer.

The system and algorithm co-optimization allows LiteGS to obtain significant superiority in both training efficiency and reconstruction quality, setting a new performance benchmark for the field. LiteGS can achieve the same quality as 3DGS-MCMC (quality SOTA) (Kheradmand et al., 2024) with up to 10.8x speedup using less than half the number of parameters. However, using the same amount of parameters, LiteGS surpasses the quality SOTA solutions by 0.2-0.4 dB in PSNR and shortens the training time by 3.8x to 7x. For lightweight models, LiteGS requires only about 10% training time and 20% parameters to produce the same quality as the original 3DGS (Kerbl et al., 2023). Compared to other cutting-edge acceleration works in lightweight models such as Mini-splatting v2 (Fang & Wang, 2024b), LiteGS also reduces the training cost by up to 1.5x. Comprehensive ablation studies further quantify and confirm the effectiveness and necessity of each technique proposed in our framework.

In summary, the main contributions of this paper are as follows.

- We identify the major bottlenecks in the existing 3DGS training pipeline, and propose an array of techniques, e.g., warp-based rasterization, Cluster-Cull-Compact, and novel densification strategies, to eliminate these bottlenecks.

- We propose a high-performance framework, LiteGS, to systematically accelerate 3DGS training through system and algorithm codesign. We have open-sourced the framework to the community.

- We perform extensive experiments and ablation studies to evaluate the effectiveness of LiteGS. Compared to the original 3DGS framework, it reduces the training time by up to 13.4x while still retaining comparable quality. In high-quality setting, it accelerates the quality SOTA solution by up to 10.8x with even slightly better quality.

## 2 Related Work

### 2.1 Acceleration

Although 3DGS is already faster to optimize than classic NeRF, the training still takes tens of minutes or even hours. A key focus in recent 3DGS research has been on removing redundant computations to speed up training. Several works reduce the per-pixel Gaussian overhead by culling or skipping negligible contributions (Liu et al., 2025; Hanson et al., 2025a; Feng et al., 2025). Hou et al. (2025) approximate the alpha blending with a commutative weighted sum, removing the need for per-frame sorting and modestly improving performance on mobile GPUs. Another line of work is pruning redundant primitives. LightGaussian (Fan et al., 2024) proposes a heuristic pruning pipeline. PUP 3D-GS (Hanson et al., 2025b) uses a Hessian-based sensitivity metric to remove 90% of Gaussians from a pretrained model while preserving detail. Other works try to improve the GPU utilizations of 3DGS training. To reduce atomic add operations in the gradient accumulation step, DISTWAR introduced an optimized reduction primitive that leverages warp-level parallelism(Durvasula & Chai, 2024). To remove atomic conflicts on the same tile, Taming 3DGS proposed a 'per-splat parallelization' approach for backpropagation (Mallick et al., 2024). TC-GS uses Tensor Core units for splatting operations, formulating alpha-blending calculations as matrix multiplies so that otherwise underutilized hardware can accelerate them (Liao et al., 2025).

### 2.2 Densification Strategies

A key to 3DGS's success is its ability to refine the point cloud during training via Adaptive Density Control (ADC) - i.e. adding Gaussians in undersampled regions and removing those with negligible opacity. However, the original ADC heuristic has limitations that often produce an explosion of Gaussians with diminishing importance. Recent works have therefore revised the densification logic to be more principled and efficient. Bulò et al. (2024) propose an error-driven densification scheme: instead of using just gradient magnitude, they compute an auxiliary per-pixel error map and spawn new Gaussians where the reconstruction error remains high. Deng et al. (2024) tackle the root cause of Gaussian overshoot by replacing the clone-and-split step with a single long-axis split that avoids overlapping the child Gaussians. Their efficient density control (EDC) method achieves the same or better fidelity using only 1/3 of the Gaussians, by preventing the creation of redundant, overlapping splats. Other strategies guide densification using additional cues. Kim et al. (2024) incorporate color residuals (via spherical harmonics gradients) into the densification criterion, which helps target new Gaussians to areas of color discrepancy rather than just geometric edges. Taming 3DGS (Mallick et al., 2024) uses a score-based selection that adds only the Gaussians which maximally improve the scene's SSIM/PSNR, and it grows the model to a predetermined budget (rather than indefinitely). By steering new Gaussians to where they raise fidelity the most and stopping when the budget is reached, this method obtains high reconstruction quality with 4-5x fewer Gaussians than the naive ADC (Kerbl et al., 2023).

## 3 Methodology

3DGS training starts with a point cloud initialized from Structure-from-Motion (SFM), followed by the optimization of primitive parameters via gradient descent. To enhance reconstruction fidelity, the framework strategically performs densification at regular intervals, augmenting the number of primitives. The Gaussian scene is represented as $\mathbb{G} = \{g_i | i = 1, \ldots, N\}$, a collection of $N$ primitives. Each primitive $g_i = \{\mathbf{p}_i^{\text{world}}, \Sigma_i^{\text{world}}, \mathbf{c}_i, o_i\}$ is defined by its 3D position in world coordinates $\mathbf{p}_{\text{world}}^i$, a 3D covariance matrix $\Sigma_{\text{world}}^i$, RGB color coefficients $\mathbf{c}_i$, and an opacity value $o_i$. The camera views are donated as $\Pi = \{\pi_i | i = 1, \ldots, M\}$ and $M$ is the number of views in the training set. The Gaussian primitive projected into screen space is represented as $\{x_{i,\pi_j}^{screen}, y_{i,\pi_j}^{screen}, \Sigma_{i,\pi_j}^{screen} \mathbf{c}_i, o_i\}$. The rendering process in 3DGS is mathematically defined as,

$$I_{\pi_j}(x,y) = \sum_{i=0}^{N'} T_{s(i),\pi_j}(x,y) \cdot \alpha_{s(i),\pi_j}(x,y) \cdot c_{s(i)} \tag{1}$$

where $s(i)$ denotes the index mapping function for depth-ordered primitives in camera space, corresponding to the $i$-th sorted primitive in the Gaussian scene representation $\mathbb{G}$, $T$ is the transmittance $T_{s(i),\pi_j}(x,y) = T_{s(i-1),\pi_j}(x,y) \cdot (1 - \alpha_{s(i-1),\pi_j}(x,y))$, $N'$ represents the number of primitives visible within the view frustum, $\alpha$ is the opacity term $\alpha_{i,\pi_j}(x,y) = o_i G(x^{screen}_{i,\pi_j} - x, y^{screen}_{i,\pi_j} - y, \Sigma^{screen}_{i,\pi_j})$ and $G$ is defined as,

$$G(\Delta x, \Delta y, \Sigma) = e^{-0.5(\Delta x, \Delta y)\Sigma(\Delta x, \Delta y)^T} \tag{2}$$

We use $\mathbb{P} = \{P_j | j = 1, ..., N_{tile}\}$ to represent the $j$-th rendering tiles, and the gradient of $c^r_{s(i)}$ is computed by Eq. (3).

$$\frac{\partial Loss}{\partial c^r_{s(i)}} = \sum_{j=0}^{N_{tile}} \sum_{(x,y) \in P_j} \frac{\partial Loss}{\partial I_r(x,y)} \cdot T_{s(i),\pi}(x,y) \cdot \alpha_{s(i),\pi}(x,y) \tag{3}$$

### 3.1 WARP-BASED RASTER

3DGS employs a tile-based method for rasterization, where a single Gaussian primitive $g_i$ might be rasterized across different pixels and tiles. Hence, the gradients of $g_i$ are computed by many threads and accumulated together during the backward pass, as shown in Eq. (3). Each primitive can have up to nine floating-point parameters, causing prohibitively high overhead in gradient reduction.

The original 3DGS implementation heavily uses global *atomic_add* operations for the reduction since the inter-tile accumulation $\sum_{j=0}^{M}$ is normally sparse. However, the accumulated intermediate pixel results in a tile, $\sum_{(x,y) \in P_j}$, is dense and therefore inefficient for atomic additions. Conventional solutions rely on warp reduction or shared memory for block-level reduction to address the performance issue, but they require the number of warp-level reductions equal to the number of warps per tile, leaving room for improvement.

To remarkably eliminate the overhead, we propose a novel *warp-based rasterization* that only requires *a single* warp reduction. The key idea is to assign only one warp to each render tile, and let each thread within the warp process multiple pixels through iterative loops. This design first performs an intra-thread reduction operation to accumulate the gradients generated by each pixel within a thread, and then applies a single warp-level reduction to aggregate the gradients for the entire tile. By shrinking the number of expensive warp reductions required per tile to just one, we substantially improve the performance of gradient reduction. The implementation details and a schematic diagram of our Warp-based Rasterization pipeline are provided in Appendix A.1.

However, the design does not come for free. It possibly leads to **sharply increased register pressure** because the approach tends to assign more pixels to a single thread, which inevitably needs to allocate more temporary variables (such as color and opacity during alpha blending) in the local register. Excessive register occupation causes register spilling, which in turn leads to notable GPU under-utilization. We propose two techniques to ease the register pressure.

#### 3.1.1 SCANLINE ALGORITHM

The first technique, inspired by the scanline algorithm(Foley et al., 1996; Shirley et al., 2015) used in software rasterization, reforms the computation of the 2D Gaussian function to maximize data reuse hence diminish register allocation in each thread. This idea lies in the observation that there is a considerable amount of redundant computations during processing multiple pixels for the same Gaussian primitive.

Along any scanline (e.g., the y-axis), the exponent of the Gaussian function (Eq. (2)) can be decomposed into a quadratic polynomial of the pixel offset $j$: Basic + Linear $\cdot j$ + Quad $\cdot j^2$. This structure allows for significant optimization: the coefficients are computed only once per scanline. The values for all subsequent pixels are then found efficiently through a few incremental multiply-add operations. The derivation of the formulas for the forward and backward phases is shown in Appendix A.2. The new 2D Gaussian scanline formulation brings two benefits. First, it effectively reduces the variables needed for each pixel calculation from 7 $(x, x^{screen}_i, y, y^{screen}_i, a, b, c)$ to 3 (basic, linear, quad). Second, it cuts the number of required multiplications (or Fused Multiply and

Add) by $G(\Delta x, \Delta y, \Sigma^{\text{screen}})$ from 9 to 2. The same principle applies to the backward pass, where gradients for different pixels with respect to the Basic, Linear, and Quad terms are first summed within the thread, saving both registers and computation.

### 3.1.2 STRATEGIC MIXED-PRECISION COMPUTING

The second technique leverages mixed-precision to improve GPU occupancy and further reduce register pressure. For alpha blending, we strategically select half-precision (FP16) instead of single-precision (FP32) for temporary variables during the blending process (e.g. Gaussian intensity $G(\Delta x, \Delta y, \Sigma^{\text{screen}})$, transmittance $T$, and color $c$). FP16 not only halves the register usage but also doubles SIMD instruction performance and the data stored in registers simultaneously, therefore delivering nearly 2x blending throughput. We chose FP16 over BF16 is because FP16 offers higher precision for the iterative transmittance calculation $T_{i+1} = T_i(1 - \alpha)$.

In addition, FP16 is applied at places that only require low precision for gradient, such as color and opacity. In these cases, we use the half2 data type to pack more gradient data for reduction, halving the number of warp reductions needed by the data represented in FP32. Instead, the covariance matrix requires high-precision gradients. We design a novel method to accelerate floating-point accumulation via hardware integer warp-reduce instructions available on major commercial GPUs, such as NVIDIA Ampere and newer generations. This method is accomplished in a four-step manner: 1) find the maximum exponent $e_{\max}$ using a single hardware instruction, *warp-reduce-max*; 2) align the mantissas of all floating-point numbers to this exponent; 3) invoke the *warp-reduce-add* instruction to sum all integerized mantissas; and 4) reconstruct the floating-point result.

These synergistic optimizations help us effectively tune the best performance given the hardware resource constraints. For example, based on the optimized per-pixel register usage, we discovered that each thread processes 4 pixels is the optimal configuration. Using the "one warp per tile" principle, we identified that tile size with $4 \times 32 = 128$ pixels yields the best performance. This size fits the major NVIDIA graphic GPUs since they have the same register file size.

### 3.2 CLUSTER-CULL-COMPACT

The scalability of 3DGS training presents a significant performance bottleneck. As training progresses, the number of Gaussian primitives can surge into millions, rendering per-primitive operations such as culling computationally prohibitive. This challenge is exacerbated by a critical architectural issue: the lack of spatial locality in the data structure. New primitives are simply appended to the end of an array, causing spatially adjacent Gaussians to be scattered across different memory locations. This disordered arrangement incurs two severe performance penalties on the GPU. First, the chaotic memory layout severely undermines cache locality, resulting in frequent cache misses that slow down parameter reads during both forward and backward passes. Second, after visibility culling, active and inactive primitives are randomly interleaved, causing severe warp divergence. This forces the entire warps to execute even if only one thread is active, wasting substantial computational resources on mostly idle operations.

To address these challenges, we adapt the established "Cluster-Cull-Compact" paradigm from the field of real-time rendering (Laine et al., 2011; Hapala et al., 2011), tailoring it for the dynamic training process of 3DGS. This pipeline consists of three stages: 1) grouping primitives into spatially contiguous clusters; 2) performing culling at the cluster granularity; and 3) compacting the data of visible clusters into a continuous memory buffer. However, traditional clustering methods like Bounding Volume Hierarchies (BVH) or graph partitioning(Karis, 2021) have high computational complexity for a training loop. The cornerstone of our approach is the utilization of Morton coding (Z-order curve) to achieve high-performance locality-aware sorting(Morton, 1966). By mapping 3D coordinates to a 1D sequence that preserves locality, Morton coding enables us to efficiently enforce spatial coherence at a minimal cost and facilitate lightweight clustering operations.

### 3.2.1 MORTON CODES

Our pipeline begins by normalizing the Gaussian coordinates $p_i = (x_i, y_i, z_i)$ into the unit cube $[0, 1]^3$. To generate a 64-bit Morton code, the normalized coordinates are scaled by $2^{21} - 1$ and converted to integers. The 64-bit Morton code is then generated by interleaving the bits of the inte-

ger coordinates: $\text{Morton}(x, y, z) = \bigoplus_{k=0}^{20} (x_k \ll (2k) \,|\, y_k \ll (2k+1) \,|\, z_k \ll (2k+2))$, where $\oplus$ denotes bit concatenation, and $x_k$ represents the $k$-th bit of the quantized coordinates.

To maintain spatial locality throughout the training, we dynamically re-sort the entire Gaussian primitive array based on their Morton codes at fixed intervals (e.g., every 5 epochs) and after each densification step. Although this encoding does not account for the varying sizes of Gaussian primitives, its extremely short execution time makes the re-sorting overhead negligible. The resulting spatially-ordered data structure is fundamental to all subsequent steps.

### 3.2.2 CLUSTER-LEVEL CULLING AND COMPACTION

Based on the Morton-sorted array, we form clusters by simply grouping contiguous primitives (e.g., 128 primitives per cluster). We then compute an axis-aligned bounding box (AABB) for each cluster. Before rendering, we perform view frustum culling at the cluster level by testing the intersection of these AABBs with the camera's view frustum, which allows us to cull large groups of invisible primitives at once. Subsequently, the data from all visible clusters are compacted into a contiguous memory buffer. This ensures that all subsequent computations benefit from coalesced memory access and reduces memory fragmentation. We provide a detailed explanation of the Morton code calculation and a visual illustration of the entire Cluster-Cull-Compact pipeline in Appendix A.3.

### 3.2.3 SYNERGY WITH SPARSE GRADIENT UPDATES

Furthermore, we leverage this clustered data structure to enhance sparse gradient update strategies (e.g., TamingGS). Instead of applying sparsity at the per-primitive level, which leads to scattered memory access during parameter updates, we apply it at the cluster granularity. This approach retains the computational savings of sparse updates while ensuring that operations like gradient computation and parameter updates are performed on contiguous memory blocks, thereby maximizing memory throughput and avoiding performance penalties.

### 3.3 OPACITY GRADIENT VARIANCE METRIC

The goal of densification is to adaptively add detail in under-reconstructed regions. The original method primarily relies on the magnitude of the view-space position gradient to identify these regions, subsequently splitting or cloning the corresponding Gaussian primitives. However, this metric suffers from a fundamental ambiguity: a large gradient may not only signal a genuine under-reconstructed area that requires more primitives but could also represent a well-sampled but simply unconverged region.

This ambiguity is exacerbated by the periodic opacity reset mechanism. Forcibly setting opacities to zero induces drastic, transient errors and gradients throughout the scene. Consequently, densification decisions made shortly after a reset are often unreliable, as it becomes difficult to distinguish whether a region has a true geometric defect or is merely in a transient state of optimization following the reset. To address this ambiguity, we introduce a new metric using the variance of the opacity gradient. Instead of merely measuring the magnitude of the gradient, we analyze its distribution across all pixels that observe the same Gaussian primitive. The intuition behind this is as follows:

- Low Variance: If the opacity gradients for a Gaussian are large and consistent in direction across all relevant pixels, it typically indicates that the model is unconverged. The optimizer has a consensus on how to adjust the opacity, and thus, no new geometry is needed.

- High Variance: If the opacity gradients vary significantly among different pixels (e.g., some pixels on a silhouette require high opacity, while others that "see through" the object require low opacity), this conflict signals a true under-reconstructed area. High variance indicates that a single Gaussian is no longer sufficient to represent the local geometric complexity, making it a prime candidate for splitting or cloning.

This variance can be estimated efficiently. For each Gaussian primitive $i$, we compute the sum of squared gradients $S_i = \sum_{\pi \in \Pi} \sum_{(x,y)} (\frac{\partial \text{Loss}(I_\pi(x,y), \text{gt}_\pi(x,y))}{\partial o^i})^2$, the sum of gradients $M_i = \sum_{\pi \in \Pi} \sum_{(x,y)} \frac{\partial \text{Loss}(I_\pi(x,y), \text{gt}_\pi(x,y))}{\partial o^i}$, and the fragment count $C_i$. The densification metric is then calculated via $C_i \cdot \text{Var} = S_i - \frac{M_i^2}{C_i}$. Our proposed warp-based rasterization technique aggregates

these pixel-level statistics with negligible overhead, enabling the efficient and low-cost computation of these moments.

Periodically resetting opacity can improve reconstruction quality by re-weighting all scene parameters, which are determined by the transmittance on each fragment. While performing more frequent resets is an intuitive idea, the disruptive impact of opacity resets adversely affects the densification process. To mitigate these effects and allow for a higher reset frequency, we replace the hard reset with a gentler "opacity decay" mechanism. Instead of setting opacities directly to zero, we decay them by half. This approach ensures that the weights of all parameters in view are reset due to the increased transmittance, while also allowing the scene to recover to normal opacity levels much more quickly. Ultimately, with the opacity decay strategy, our densification hyperparameters are adjusted accordingly: the densification interval is extended to 5 epochs, which provides sufficient time for the scene to reach a "quasi-stable state" after an opacity adjustment for reliable variance metric calculation, while the opacity decay itself is performed every 10 epochs.

## 4 EXPERIMENTS

We conducted extensive experiments to validate the effectiveness of our proposed framework.

**Datasets**: We evaluated LiteGS on three datasets: Mip-NeRF 360 (Barron et al., 2022), Tank & Temples (Knapitsch et al., 2017), and Deep Blending (Hedman et al., 2018). For Mip-NeRF 360, we uniformly use the pre-downsampled images provided with the official dataset for all methods.

**Baselines**: We compare LiteGS against a comprehensive set of baselines, including foundational methods like the original 3DGS (Kerbl et al., 2023), high-quality reconstruction methods like 3DGS-MCMC (Kheradmand et al., 2024), SSS (Zhu et al., 2025), and recent acceleration-focused works such as TamingGS (Mallick et al., 2024), DashGaussian (Chen et al., 2025), Mini-Splatting (Fang & Wang, 2024a), and Mini-Splatting v2 (Fang & Wang, 2024b).

**Implementation Details**: All experiments are performed on a single NVIDIA RTX 3090 GPU. PSNR, SSIM, and LPIPS are used as evaluation metrics. LiteGS adopts the same learning rates as TamingGS. Our densification starts after the third epoch, with splitting operations performed every five epochs. Opacity decay is applied every 10 epochs during the first 80% of training.

**Parameter Scale**: We carried out experiments on three distinct parameter scales. *LiteGS-turbo* runs at a small parameter scale and the number of training iterations (18K) is identical to Mini-splatting v2, aiming to showcase its high training performance. *LiteGS-balance* selects a moderate number of parameters to strike a balance between training accuracy and speed, using 30K training iterations. *LiteGS-quality* uses a parameter count comparable to the original 3DGS to achieve the highest possible reconstruction quality with also 30K training iterations. In addition to these three typical parameter scales, we provide more comparative results across a broader range of parameter scales in Appendix B.1.

### 4.1 COMPARISON WITH STATE-OF-THE-ART METHODS

Table 1 presents the aggregate results from our comparative experiments with other leading methods, while the detailed per-scene metrics and comprehensive statistical analysis are provided in Appendix B.2. Qualitative visual comparisons demonstrating the reconstruction quality differences are available in Appendix B.3. Our framework demonstrates dominant and Pareto-optimal performance across all scales, achieving superior results in both rapid and high-quality reconstruction tasks. The metrics for 3DGS-MCMC and SSS in Table 1 differ from their published results due to their use of non-standard preprocessing on the Mip-NeRF 360 dataset, whereas Table 1 uses a unified, standard preprocessing for all methods. For the sake of a fair comparison, we also report results using their non-standard preprocessing pipeline in Appendix B.4.

**Rapid Reconstruction**. In the lightweight setting, LiteGS-turbo establishes a new SOTA for speed-quality trade-off. As shown in Table 1, compared to the original 3DGS, LiteGS-turbo reduces the number of primitives/parameters and the training time by up to 5.8x and 13.4x, respectively. It is also 1.5x faster than the currently fastest lightweight method, Mini-Splatting v2, with comparable accuracy (slightly better PSNR but worse LPIPS, as fewer points capture less high-frequency information). Another key advantage of LiteGS over Mini-splatting lies in the slow and predictable

Table 1: Comparison of SSIM, PSNR, LPIPS, training time, and number of Gaussians across three datasets. * means the training iteration is 40K. For a fair comparison, LiteGS uses the same number of iterations as SSS. LiteGS-quality gets 0.870/24.72/0.133 (better than others except SSS) in 30K iterations and takes 398s.

| Method | Mip-NeRF 360 | | | Tanks&Temples | | | Deep Blending | | |
|---|---|---|---|---|---|---|---|---|---|
| | SSIM/PSNR/LPIPS | Train(s) | Num | SSIM/PSNR/LPIPS | Train(s) | Num | SSIM/PSNR/LPIPS | Train(s) | Num |
| TamingGS-budget | 0.801 / 27.32 / 0.257 | 380 | 0.68 | 0.834 / 23.73 / 0.210 | 238 | 0.31 | 0.904 / 29.90 / 0.255 | 294 | 0.25 |
| Mini-Splatting | **0.822** / 27.32 / 0.217 | 1221 | 0.49 | **0.846** / 23.43 / **0.180** | 755 | 0.30 | 0.910 / 29.98 / **0.240** | 1036 | 0.56 |
| Mini-Splatting v2 | 0.821 / 27.33 / **0.215** | 214 | 0.62 | 0.841 / 23.14 / 0.186 | 142 | 0.35 | **0.912** / 30.08 / 0.240 | 165 | 0.65 |
| LiteGS-turbo | 0.810 / **27.70** / 0.236 | **145** | 0.58 | 0.835 / **23.84** / 0.201 | **108** | 0.35 | 0.911 / **30.42** / 0.248 | **111** | 0.65 |
| 3DGS | 0.815 / 27.47 / 0.216 | 1626 | 3.35 | 0.848 / 23.66 / 0.176 | 906 | 1.84 | 0.904 / 29.54 / 0.244 | 1491 | 2.82 |
| DashGaussian | 0.816 / 27.66 / 0.220 | 412 | 2.23 | 0.851 / 24.02 / 0.180 | 318 | 1.20 | 0.908 / 30.14 / 0.247 | 285 | 1.91 |
| TamingGS-big | 0.822 / 27.84 / 0.207 | 906 | 3.30 | 0.855 / 24.17 / 0.167 | 547 | 1.83 | 0.908 / 29.88 / 0.234 | 813 | 2.80 |
| 3DGS-MCMC | 0.834 / 28.08 / 0.186 | 3142 | 3.21 | 0.860 / 24.29 / 0.190 | 1506 | 1.85 | 0.890 / 29.67 / 0.320 | 2319 | 2.95 |
| SSS | 0.828 / 27.92 / 0.187 | 3961 | 2.31 | **0.873\*** / 24.87\* / **0.138\*** | 3067\* | 1.85 | 0.907 / 30.07 / 0.247 | 1772 | 0.60 |
| LiteGS-balance | 0.827 / 28.13 / 0.205 | **301** | 1.10 | 0.863 / 24.61 / 0.151 | **263** | 0.90 | **0.912** / **30.37** / 0.233 | **215** | 1.00 |
| LiteGS-quality | **0.836** / **28.25** / **0.176** | 515 | 3.30 | **0.873\*** / **24.91\*** / **0.133\*** | 531\* | 1.83 | **0.912** / 30.34 / **0.218** | 330 | 2.80 |

Table 2: Ablation study on the effect of our Cluster-Cull-Compact pipeline. Disabling cluster-level culling (w/o culling) significantly increases training time with negligible impact on final image quality across all datasets.

| | Mip-NeRF 360 | | Tanks&Temples | | Deep Blending | |
|---|---|---|---|---|---|---|
| | SSIM/PSNR/LPIPS | Train(s) | SSIM/PSNR/LPIPS | Train(s) | SSIM/PSNR/LPIPS | Train(s) |
| w/ culling | 0.836 / 28.25 / 0.176 | 515 | 0.870 / 24.72 / 0.133 | 398 | 0.912 / 30.34 / 0.218 | 330 |
| w/o culling | 0.834 / 28.28 / 0.176 | 701 | 0.870 / 24.75 / 0.133 | 483 | 0.912 / 30.29 / 0.217 | 533 |

parameter expansion, but the latter uses an aggressive densify-and-prune strategy for lightweighting. Hence, LiteGS has a lower peak memory footprint, offering a significant advantage on devices with limited memory budgets.

**High-Quality Reconstruction**. In the million-scale parameter regime, LiteGS significantly outperforms its counterparts. LiteGS-balance, using only $1/3$ to $1/2$ the parameters of 3DGS-MCMC, is on par with its reconstruction quality but reducing the training time by an order of magnitude, i.e., from nearly an hour to 3-5 minutes. Even compared to the SOTA acceleration framework, TamingGS, LiteGS-balance cuts the training time by up to 3.7x (with an average of 3x), plus delivers significantly higher accuracy. DashGaussian[1], by reducing the number of primitives, shows a slight advantage in training speed over TamingGS, but suffers a significant drop in accuracy. Furthermore, LiteGS-quality sets a new SOTA for quality. Compared to 3DGS-MCMC, it not only speeds up training performance by 3.8x-7x (about 5-9 minutes), but also improves PSRN by $0.2$-$0.4$ dB.

These results indicate that our high-performance framework, LiteGS, is able to deliver: 1) minute-level 3DGS training for the small parameter count scenarios without comprising reconstruction quality, 2) extremely high quality for the large parameter count tasks with remarkably less training overhead compared to virtually all existing solutions.

## 4.2 ABLATION STUDIES

This section validates the effectiveness of each key component of LiteGS through ablation studies.

**Impact of Cluster-Cull-Compact**. To isolate its effect, we disable our spatial clustering pipeline in the LiteGS-quality setting. The results in Table 2 show that it significantly increases training time (e.g. 36% slower, from 515s to 701s on Mip-NeRF 360) with only a minor impact on quality metrics. This reveals that our Cluster-Cull-Compact module effectively accelerates training.

**Impact of Warp-based Rasterizer**. An end-to-end training time comparison is insufficient to fairly evaluate rasterizer performance, as our densification strategy may generate more visible primitives.

---
[1]Metrics are sourced from their GitHub

Table 3: Performance comparison of different rasterization methods on the 'garden' scene with varying numbers of primitives (500k, 2000k, 5728k). Our Warp-based Rasterizer shows a significant speedup in both forward and backward passes. Atomic Raster is the original implementaion in 3DGS. Per-splat Raster is the implementation from TamingGS which may be the most popular gaussian raster operator. Tensor-core GS (Liao et al., 2025) is the recent SOTA work for 3DGS inference and Warp-based Raster is our work.

| | Forward(ms) | | | Backward(ms) | | |
|---|---|---|---|---|---|---|
| | 500k | 2000k | 5728k | 500k | 2000k | 5728k |
| Atomic | 5.45 | 10.56 | 17.82 | 25.26 | 37.23 | 51.84 |
| Per-splat | 5.06 | 7.49 | 12.58 | 7.53 | 13.46 | 27.31 |
| Tensor-core | 1.79 | 3.59 | 6.81 | - | - | - |
| Warp-based | **0.36** | **1.45** | **3.01** | **1.87** | **3.64** | **6.93** |

Therefore, we list the cost breakdown of the operators in Table 3. Our warp-based rasterizer demonstrates a prominent speedup across all primitive scales, outperforming the original 3DGS atomic rasterizer by 7.0x-13.0x, popular per-splat rasterizers by 4.0x, and even recent SOTA methods (Liao et al., 2025) using Tensor Cores by 2x.

**Impact of Densification Strategy**. Table 4 depicts the performance of each component in our densification strategy. In the setting without opacity decay, we use the hard opacity reset scheme and the same hyperparameters in the original 3DGS. In the setting without opacity gradient variance metric, per-pixel error (Bulò et al., 2024) is taken as the densification indicator. Using the original 3DGS densification hyperparameters and opacity reset policy could drop PSNR by 0.3-0.4 dB. This suggests that a more frequent reduction in overall scene opacity improves reconstruction quality, and our opacity decay mechanism avoids the instability of consecutive hard resets by allowing faster recovery. In the control experiment using per-pixel error, the PSNR drops by approx. 0.3-0.6 dB, indicating that the opacity gradient variance is a more robust metric than the error-based indicator.

Table 4: Ablation study on our densification strategy. We compare our full model against variants without opacity decay (w/o Decay), without the gradient variance metric (w/o Var), and without both (w/o Both).

| | Mip-NeRF 360 | | | Tanks & Temples | | | Deep Blending | | |
|---|---|---|---|---|---|---|---|---|---|
| | PSNR | SSIM | LPIPS | PSNR | SSIM | LPIPS | PSNR | SSIM | LPIPS |
| LiteGS | **28.12** | **0.824** | **0.209** | **24.52** | **0.866** | **0.145** | **30.37** | **0.912** | **0.233** |
| w/o Decay | 27.68 | 0.813 | 0.222 | 24.11 | 0.852 | 0.166 | 29.93 | 0.908 | 0.239 |
| w/o Var | 27.40 | 0.813 | 0.235 | 24.27 | 0.853 | 0.163 | 30.08 | 0.905 | 0.245 |
| w/o Both | 27.13 | 0.777 | 0.259 | 23.97 | 0.837 | 0.190 | 29.76 | 0.900 | 0.259 |

## 5 CONCLUSION

This paper addresses the critical problem of the training efficiency bottleneck in 3D Gaussian Splatting (3DGS), which severely limits its potential in applications. We conducted an in-depth analysis and identified that this bottleneck was not caused by a single factor, but rather stemmed from systemic, interconnected design flaws at three levels: top-level algorithm design, mid-level data structures, and low-level GPU computation. To address this, we proposed LiteGS, a layered, synergistic optimization framework that systematically enhances the training pipeline. Extensive experiments compellingly demonstrated the superiority of our framework. LiteGS reduces training time by an order of magnitude while achieving reconstruction quality that is comparable to or even surpasses existing state-of-the-art methods across various parameter scales, setting a new SOTA performance benchmark for the field. Although LiteGS has achieved significant success, there remain avenues for future exploration. Our current Morton-based spatial sorting primarily considers Gaussian centers; future work could investigate data structures that simultaneously account for primitive size and anisotropy. Furthermore, extending our efficient framework to the real-time capture and rendering of dynamic scenes, as well as further lightweighting for mobile and edge devices, are highly valuable research directions. The authors wish to acknowledge the use of a Large Language Model (LLM) to assist with refining the grammar of this manuscript.

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

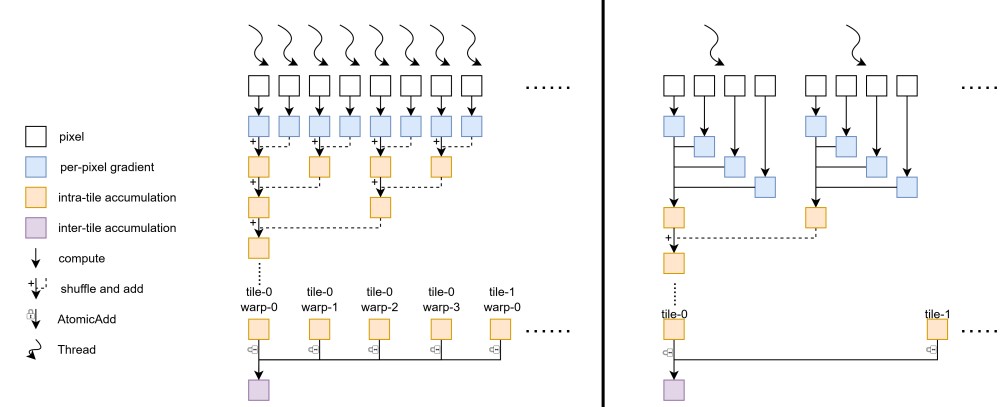

Figure 2: Comparison of gradient computation pipelines. The left side illustrates a conventional approach where per-pixel gradients are computed, followed by warp reduction. The right side depicts our proposed Warp-based Rasterization strategy, which employs a more efficient mapping: warps handle tiles. This design reduces redundant operations by requiring only one warp-level shuffle-reduction and one atomic addition per tile per gradient, significantly optimizing the computation flow.

## A    DETAIL OF OUR METHODS

### A.1    DETAIL OF WARP-BASED RASTER

The gradient computation in 3DGS can be decomposed into three primary components: (1) per-pixel gradient computation $f_{\text{pixel}}$; (2) intra-tile accumulation $f_{\text{tile}}$; and (3) inter-tile accumulation $f_{\text{global}}$. Taking $c_{s(i)}^r$ as an example, the per-pixel gradient computation can be expressed as Eq. (4), intra-tile accumulation as Eq. (5), and inter-tile accumulation as Eq. (6).

$$f_{\text{pixel}}(x, y) = \frac{\partial Loss}{\partial I_r(x, y)} \cdot T_{s(i), \pi}(x, y) \cdot \alpha_{s(i), \pi}(x, y) \tag{4}$$

$$f_{\text{tile}}(P) = \sum_{(x,y) \in P} f_p(x, y) \tag{5}$$

$$f_{\text{global}} = \sum_{P \in \mathbb{P}} f_t(P) \tag{6}$$

Owing to the property of 2D Gaussians where their values diminish nearly to zero at coordinates distant from the mean, the original 3DGS algorithm incorporates tile-level culling. This process eliminates tiles whose intensity has decayed to almost zero from Eq. (6), thereby transforming the inter-tile accumulation into a sparse reduction operation. Consequently, employing atomic operations for this summation is an appropriate approach.

However, the intra-tile accumulation stage is where performance bottlenecks predominantly manifest. Handling both inter-tile and intra-tile accumulation together uniformly using atomic operations is, from both a performance and hardware perspective, a crude and inefficient practice. While employing warp-level or block-level reduction techniques represents a common improvement, redundancy persists between atomic operations and reduction schemes. As illustrated in Fig. 2, our Warp-based Rasterization proposes a more rational mapping between hardware units and algorithmic levels: thread-to-pixels and warp-to-tile. Specifically, individual threads are responsible for computing gradients for a set of pixels and summing them within the thread. An intra-tile accumulation is then achieved through a warp-level shuffle reduction operation. Finally, a single atomic addition per gradient per tile is issued to complete the inter-tile accumulation.

---

**Algorithm 1** Scanline algorithm

---

**Input**: Number of visible primitives $N$, Depth sorted primitives $G = \{g_i | i = 1, 2, 3, ...N\}$, $g_i = \{x_i^{screen}, y_i^{screen}, c_i, \Sigma_i^{screen}, o_i\}$, Coordinate of first pixel in scanline $(x, y)$
**Parameter**: $length$
**Output**: $out\_color \in \mathbb{R}^{length \times 3}$

1: $T \in \mathbb{R}^{length} \leftarrow 1.0$
2: **for** $i \leftarrow 0$ to $N$ **do**
3:    $\Delta x = x_i^{screen} - x$
4:    $\Delta y = y_i^{screen} - y$
5:    $basic = -0.5(\Delta x, \Delta y)\Sigma_i^{screen}(\Delta x, \Delta y)^T$
6:    $linear = b\Delta x + c\Delta y$
7:    $quad = -0.5c$
8:    **for** $j \leftarrow 0$ to $length$ **do**
9:       $g = exp(basic + linear * j + quad * j^2)$
10:       $alpha = g * o^i$
11:       $out\_color[j] + = T[j] * alpha * c_i$
12:       $T[j] * = (1 - alpha)$
13:    **end for**
14: **end for**

---

The performance acceleration afforded by Warp-based Rasterization stems primarily from two factors: (1) a reduced number of warp-level reduction operations, and (2) a reduced number of atomic operations. Assuming a tile contains $32 \times N$ pixels, a naive implementation using warp-reduction for intra-tile accumulation would require $N$ warp-reduction operations and $N$ atomic additions. In contrast, our Warp-based Raster approach necessitates only **one** warp-reduction and **one** atomic addition per tile per gradient.

## A.2 DETAIL OF GS SCANLINE ALGORITHM

In software rasterization, the scanline algorithm serves as an effective optimization. It enables the reuse of intermediate computation results and parameters across pixels on the same scanline. For Gaussian primitives, consider the computation of 2D Gaussian rasterization in Eq. (2), where $\Sigma^{screen} = \begin{pmatrix} a & b \\ b & c \end{pmatrix}$. Eq. (2) can be expanded into Eq. (7).

$$G(\Delta x, \Delta y, \Sigma) = e^{-0.5(a\Delta x^2 + 2b\Delta x \Delta y + c\Delta y^2)} \tag{7}$$

Considering the $i$-th point $(x, y + i)$ on the scanline, $G(x^{screen} - x, y^{screen} - (y + i), \Sigma^{screen})$ can be expressed as Eq. (8).

$$G(\Delta x, \Delta y - i, \Sigma) = e^{-0.5(a\Delta x^2 + 2b\Delta x \Delta y + c\Delta y^2) + (b\Delta x + c\Delta y)i - 0.5ci^2} \tag{8}$$

By defining Basic $= -0.5(a\Delta x^2 + 2b\Delta x \Delta y + c\Delta y^2)$, Linear $= (b\Delta x + c\Delta y)$, and Quad $= -0.5c$, we obtain the scanline computation formula for Gaussian primitive software rasterization in Eq. (9). The Gaussian scanline algorithm is illustrated in Algorithm 1.

$$G(\Delta x, \Delta y - i, \Sigma) = e^{\text{Basic} + \text{Linear} \cdot i + \text{Quad} \cdot i^2} \tag{9}$$

As shown in Eq. (7), evaluating the exponential typically requires 9 floating-point instructions (assuming most additions are fused into multiplications as Fused Multiply-Add instructions, thus primarily counting multiplications). Using the method in Eq. (9), computing Basic requires 9 multiplications, Linear requires 1 addition (reusing results from Basic, though additions may not be fused), and Quad requires 1 multiplication. Given precomputed Basic, Linear, and Quad, each pixel on the scanline requires an additional 2 floating-point instructions (considering $i$ and $i^2$ as compile-time constants). Assuming a scanline length of $L$, the Gaussian scanline algorithm thus uses $11 + 2L$ floating-point instructions per primitive. When $L = 4$, the scanline algorithm uses 19 instructions,

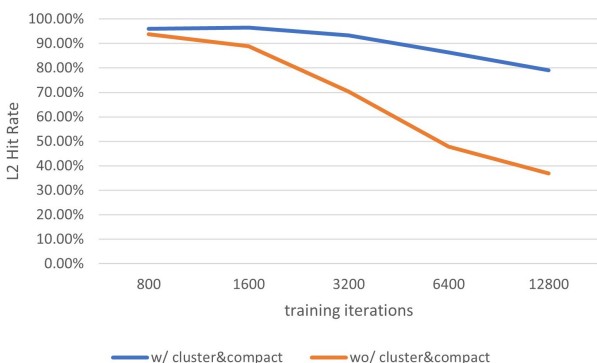

Figure 3: The evolution of L2 cache hit rate throughout training. The proposed cluster-and-compact method (blue) maintains significantly higher cache efficiency compared to the baseline without this optimization (orange), as the model grows and primitives are appended.

compared to approximately $9 \times 4 = 36$ instructions for the naive per-pixel method, which is nearly double that number.

The warp-based rasterization approach for gradient summation also benefits from the Gaussian scanline algorithm. For parameter $c$, let $l = \sum_{i=0}^{L} \frac{\partial Loss}{\partial I(x,y+i)} \frac{\partial I(x,y+i)}{\partial c}$. The per-pixel gradient computation is shown in Eq. (10). The sum of gradients along the scanline is expressed in Eq. (11).

$$
\begin{aligned}
l &= \sum_{i=0}^{L} \frac{\partial Loss}{\partial I(x,y+i)} \frac{\partial I(x,y+i)}{\partial G(\Delta x, \Delta y - i, \Sigma)} \frac{\partial G(\Delta x, \Delta y - i, \Sigma)}{\partial c} \\
&= \sum_{i=0}^{L} \frac{\partial Loss}{\partial I(x,y+i)} \frac{\partial I(x,y+i)}{\partial G(\Delta x, \Delta y - i, \Sigma)} G(\Delta x, \Delta y - i, \Sigma) \cdot (-0.5\Delta y^2 + \Delta yi - 0.5i^2)
\end{aligned}
\tag{10}
$$

$$
\begin{aligned}
l &= \frac{\partial \text{Basic}}{\partial c} \sum_{i=0}^{L} \frac{\partial Loss}{\partial I(x,y+i)} \frac{\partial I(x,y+i)}{\partial G(\Delta x, \Delta y - i, \Sigma)} \frac{\partial G(\Delta x, \Delta y - i, \Sigma)}{\partial \text{Basic}} \\
&+ \frac{\partial \text{Linear}}{\partial c} \sum_{i=0}^{L} \frac{\partial Loss}{\partial I(x,y+i)} \frac{\partial I(x,y+i)}{\partial G(\Delta x, \Delta y - i, \Sigma)} \frac{\partial G(\Delta x, \Delta y - i, \Sigma)}{\partial \text{Linear}} \\
&+ \frac{\partial \text{Quad}}{\partial c} \sum_{i=0}^{L} \frac{\partial Loss}{\partial I(x,y+i)} \frac{\partial I(x,y+i)}{\partial G(\Delta x, \Delta y - i, \Sigma)} \frac{\partial G(\Delta x, \Delta y - i, \Sigma)}{\partial \text{Quad}}
\end{aligned}
\tag{11}
$$

### A.3 DETAIL OF CLUSTER-CULL-COMPACT

The 3DGS training phase mainly faces two major issues: (1) **cache misses** caused by the unordered storage of scene primitives, and (2) **hardware inefficiency** issues introduced by view frustum culling. Fig. 3 illustrates the change in the GPU L2 cache hit rate throughout the training process as the model gradually expands. The figure clearly shows that as training progresses, new parameters are appended to the end of the memory space, degrading the coherence between the 3D spatial structure and the memory layout. Consequently, the L2 cache hit rate progressively decreases during training.

Fig. 4 demonstrates the computational and I/O efficiency problems caused by culling. As primitives lose the coherence between their 3D spatial location and their memory location, culling operations can occur randomly at any location within the memory space. Since GPU computation is scheduled in warps (32 threads), primitive-related computations are only skipped if all 32 consecutively aligned primitives within a warp are culled. In the vast majority of cases, culled primitives do not actually reduce the number of instructions executed. A similar issue occurs during parameter updates

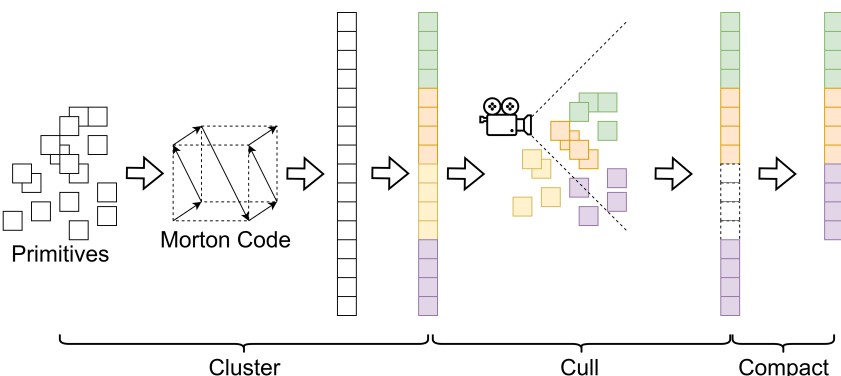

Figure 4: Illustration of GPU inefficiency caused by culling. Primitives are stored unordered in memory and view frustum culling removes primitives arbitrarily. This leads to warp divergence, as well as inefficient memory operations during the parameter update with sparse gradients.

Figure 5: The pipeline of cluster-cull-compact. Primitives are sorted according to their Morton codes. Every 128 primitives form a cluster, and an axis-aligned bounding box (AABB) is constructed for each cluster. Visibility testing and culling are performed at the cluster level, and the remaining visible clusters are then compacted into contiguous memory space.

when using sparse gradients. Global memory load/store operations on the GPU are performed at a minimum granularity of a 32-byte sector. When threads attempt to write to several noncontiguous addresses, this often results in the hardware storing multiple entire sectors. In this scenario, sparse gradients do not truly reduce the amount of data written to memory.

To address the above issues, we propose the cluster-cull-compact method, as illustrated in Fig. 5.

## B  ADDITIONAL EXPERIMENT RESULTS

### B.1  PARAMETER SCALE RESULTS

In Table 1, we only showcase experiments for three parameter scales (corresponding to turbo, balance, and quality). To further demonstrate the superiority of using the opacity gradient variance as a densification metric, we conducted additional experiments across a broader range of parameter scales, with the results visualized in Fig. 6. It is evident from the figure that when using this densification strategy, multiple scenes achieve accuracy comparable to MCMC with significantly fewer parameters.

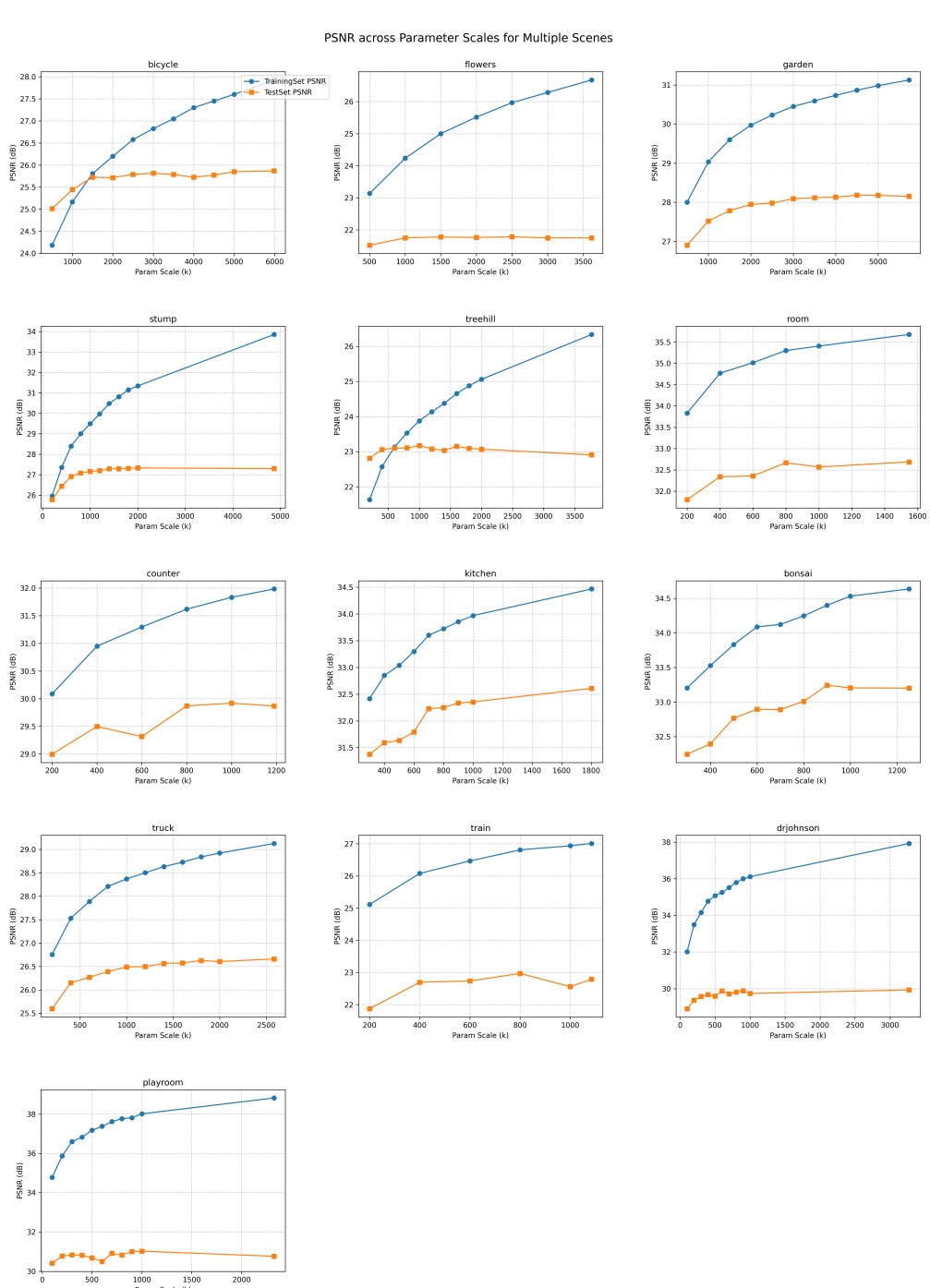

Figure 6: Comparison of training and test PSNR for LiteGS across various parameter scales and scenes. Blue lines indicate training set PSNR, while orange lines represent test set PSNR.

## B.2 PER-SCENE QUANTITATIVE RESULTS

The detailed per-scene experimental results are presented in Table 5, from which we draw the following conclusions:

- LiteGS-turbo shows accuracy on par with 3DGS and Mini-Splatting v2 (slightly better PSNR, slightly lower SSIM and LPIPS). In terms of speed, it is 10x faster than 3DGS and 1.4x faster than the current lightweight state-of-the-art method, Mini-Splatting v2, across multiple scenes.

- LiteGS-balance demonstrates accuracy comparable to 3DGS-MCMC in most scenes (with slightly better PSNR but slightly lower SSIM and LPIPS), while achieving a 10x speedup.

- LiteGS-quality significantly outperforms 3DGS-MCMC in most scenarios, while keeping its training time under 10 minutes.

## B.3 QUALITATIVE RESULTS

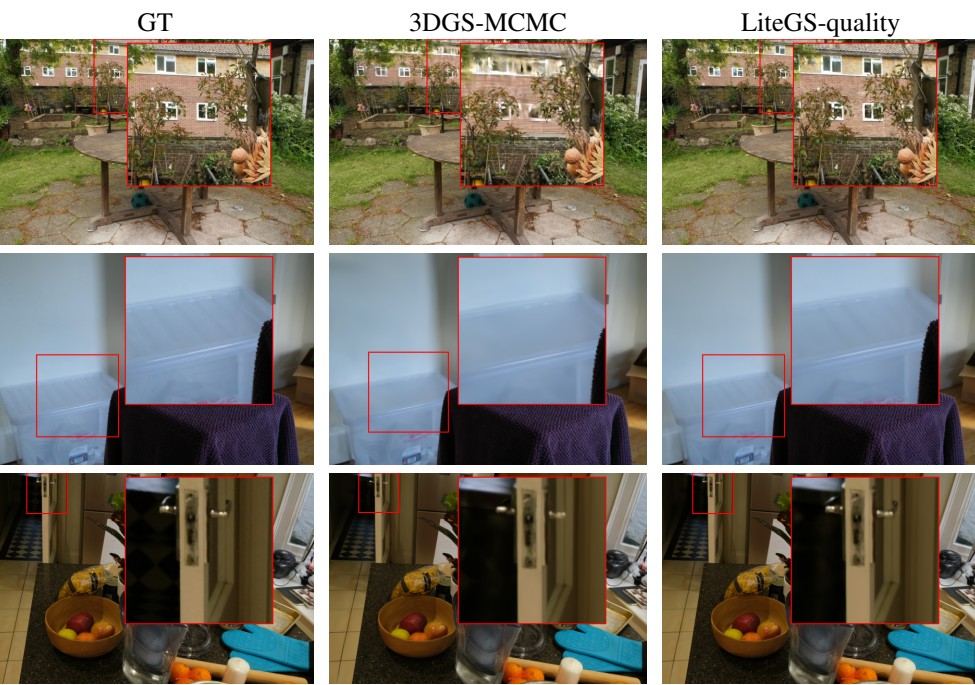

Figure 7: Qualitative comparisons between LiteGS-quality and 3DGS-MCMC. GT denotes ground truth.

To further complement the quantitative results, we provide additional qualitative comparisons between LiteGS-quality and 3DGS-MCMC in Fig. 7. The selected views are sampled from the Garden, Bonsai, and Counter scenes of the Mip-NeRF 360 (Barron et al., 2022) dataset.

Despite being trained more than 6x faster, LiteGS-quality delivers comparable or even superior visual quality. In the Garden scene, LiteGS-quality reconstructs finer branch structures and avoids oversmoothing. In Bonsai, LiteGS-quality captures sharper planar boundaries with fewer artifacts. For Counter, LiteGS-quality preserves edge contrast and color consistency, while 3DGS-MCMC exhibits visible blur and shading drift.

These results further demonstrate the effectiveness of LiteGS in achieving high-quality reconstruction at significantly reduced training time.

Table 5: Quantitative evaluation of LiteGS and previous works.

| | bicycle | flowers | garden | stump | treehill | room | counter | kitchen | bonsai | truck | train | johnson | playroom |
|---|---|---|---|---|---|---|---|---|---|---|---|---|---|
| **SSIM ↑** | | | | | | | | | | | | | |
| TamingGS-budget | 0.765 | 0.556 | 0.858 | 0.739 | 0.628 | 0.908 | 0.898 | 0.922 | 0.935 | 0.867 | 0.802 | 0.902 | 0.906 |
| Mini-Splatting | 0.773 | 0.626 | 0.847 | 0.806 | 0.652 | 0.921 | 0.905 | 0.926 | 0.939 | 0.882 | 0.810 | 0.907 | 0.912 |
| Mini-Splatting v2 | 0.771 | 0.612 | 0.850 | 0.797 | 0.651 | 0.922 | 0.908 | 0.929 | 0.944 | 0.875 | 0.806 | 0.910 | 0.914 |
| LiteGS-turbo | 0.744 | 0.597 | 0.843 | 0.783 | 0.629 | 0.920 | 0.906 | 0.924 | 0.944 | 0.876 | 0.794 | 0.908 | 0.915 |
| 3DGS | 0.765 | 0.606 | 0.866 | 0.773 | 0.633 | 0.919 | 0.909 | 0.928 | 0.942 | 0.882 | 0.814 | 0.901 | 0.907 |
| TamingGS-big | 0.779 | 0.613 | 0.873 | 0.787 | 0.646 | 0.923 | 0.909 | 0.931 | 0.942 | 0.891 | 0.819 | 0.907 | 0.908 |
| 3DGS-MCMC | 0.787 | 0.634 | 0.868 | 0.811 | 0.661 | 0.937 | 0.917 | 0.937 | 0.948 | 0.900 | 0.841 | 0.903 | 0.915 |
| SSS | 0.784 | 0.629 | 0.873 | 0.801 | 0.640 | 0.930 | 0.918 | 0.934 | 0.951 | 0.896 | 0.850 | 0.907 | 0.915 |
| LiteGS-balance | 0.787 | 0.620 | 0.872 | 0.800 | 0.644 | 0.930 | 0.911 | 0.933 | 0.948 | 0.897 | 0.829 | 0.910 | 0.915 |
| LiteGS-quality | 0.798 | 0.646 | 0.881 | 0.804 | 0.652 | 0.933 | 0.922 | 0.936 | 0.952 | 0.901 | 0.839 | 0.910 | 0.914 |
| **PSNR ↑** | | | | | | | | | | | | | |
| TamingGS-budget | 25.19 | 21.09 | 27.40 | 26.15 | 23.01 | 31.37 | 28.60 | 31.13 | 31.85 | 25.19 | 22.27 | 29.45 | 30.36 |
| Mini-Splatting | 25.21 | 21.58 | 26.82 | 27.19 | 22.63 | 31.17 | 28.57 | 31.25 | 31.48 | 25.33 | 21.53 | 29.50 | 30.46 |
| Mini-Splatting v2 | 25.25 | 21.37 | 26.80 | 27.10 | 22.76 | 31.24 | 28.59 | 31.29 | 31.72 | 24.97 | 21.29 | 29.60 | 30.47 |
| LiteGS-turbo | 25.09 | 21.65 | 27.27 | 27.11 | 23.09 | 31.96 | 29.07 | 31.66 | 32.44 | 25.80 | 21.89 | 29.74 | 31.11 |
| 3DGS | 25.19 | 21.57 | 27.39 | 26.60 | 22.53 | 31.47 | 29.07 | 31.51 | 32.07 | 25.38 | 22.04 | 29.09 | 29.98 |
| TamingGS-big | 25.49 | 21.84 | 27.76 | 27.03 | 22.98 | 32.18 | 29.02 | 31.88 | 32.38 | 25.90 | 22.44 | 29.55 | 30.21 |
| 3DGS-MCMC | 25.65 | 22.05 | 27.78 | 27.36 | 22.97 | 32.09 | 29.34 | 32.04 | 32.65 | 26.47 | 22.54 | 29.45 | 30.36 |
| SSS | 25.22 | 21.21 | 27.48 | 26.87 | 22.39 | 32.23 | 29.69 | 32.29 | 33.46 | 26.40 | 23.07 | 29.78 | 30.48 |
| LiteGS-balance | 25.71 | 21.75 | 27.95 | 27.28 | 23.11 | 32.67 | 29.50 | 32.33 | 32.90 | 26.50 | 22.74 | 29.73 | 31.02 |
| LiteGS-quality | 25.86 | 21.75 | 28.15 | 27.29 | 22.91 | 32.69 | 29.87 | 32.61 | 33.20 | 26.66 | 22.79 | 29.93 | 30.76 |
| **LPIPS ↓** | | | | | | | | | | | | | |
| TamingGS-budget | 0.284 | 0.406 | 0.126 | 0.288 | 0.380 | 0.249 | 0.222 | 0.140 | 0.219 | 0.184 | 0.237 | 0.265 | 0.245 |
| Mini-Splatting | 0.314 | 0.327 | 0.150 | 0.198 | 0.314 | 0.213 | 0.198 | 0.129 | 0.200 | 0.139 | 0.222 | 0.244 | 0.239 |
| Mini-Splatting v2 | 0.321 | 0.331 | 0.142 | 0.205 | 0.321 | 0.207 | 0.193 | 0.125 | 0.190 | 0.150 | 0.222 | 0.238 | 0.242 |
| LiteGS-turbo | 0.261 | 0.353 | 0.155 | 0.235 | 0.366 | 0.220 | 0.205 | 0.133 | 0.198 | 0.159 | 0.243 | 0.251 | 0.245 |
| 3DGS | 0.211 | 0.336 | 0.107 | 0.215 | 0.324 | 0.219 | 0.200 | 0.126 | 0.203 | 0.147 | 0.207 | 0.245 | 0.244 |
| TamingGS-big | 0.193 | 0.332 | 0.099 | 0.196 | 0.314 | 0.209 | 0.198 | 0.122 | 0.201 | 0.128 | 0.207 | 0.233 | 0.235 |
| 3DGS-MCMC | 0.169 | 0.282 | 0.095 | 0.170 | 0.272 | 0.198 | 0.184 | 0.120 | 0.190 | 0.110 | 0.181 | 0.234 | 0.236 |
| SSS | 0.180 | 0.276 | 0.097 | 0.181 | 0.286 | 0.194 | 0.177 | 0.116 | 0.182 | 0.108 | 0.165 | 0.245 | 0.241 |
| LiteGS-balance | 0.188 | 0.314 | 0.106 | 0.197 | 0.333 | 0.199 | 0.198 | 0.119 | 0.195 | 0.108 | 0.194 | 0.238 | 0.229 |
| LiteGS-quality | 0.156 | 0.254 | 0.087 | 0.173 | 0.260 | 0.189 | 0.174 | 0.113 | 0.182 | 0.093 | 0.175 | 0.223 | 0.214 |
| **Primitives Number (k)** | | | | | | | | | | | | | |
| TamingGS-budget | 814 | 575 | 2081 | 480 | 785 | 225 | 311 | 482 | 413 | 272 | 365 | 404 | 185 |
| Mini-Splatting | 530 | 570 | 560 | 610 | 570 | 390 | 410 | 430 | 360 | 320 | 280 | 600 | 510 |
| Mini-Splatting v2 | 680 | 610 | 730 | 670 | 580 | 520 | 550 | 600 | 620 | 340 | 360 | 800 | 490 |
| LiteGS-turbo | 680 | 610 | 730 | 670 | 580 | 400 | 400 | 600 | 600 | 340 | 360 | 800 | 490 |
| 3DGS | 6100 | 3630 | 5840 | 4790 | 3890 | 1550 | 1200 | 1810 | 1260 | 2600 | 1090 | 3310 | 2320 |
| TamingGS-big | 5987 | 3618 | 5728 | 4867 | 3770 | 1548 | 1190 | 1803 | 1252 | 2584 | 1085 | 3273 | 2326 |
| 3DGS-MCMC | 5900 | 3600 | 5200 | 4750 | 3700 | 1500 | 1200 | 1800 | 1300 | 2600 | 1100 | 3400 | 2500 |
| SSS | 3000 | 3000 | 3000 | 3000 | 3000 | 1500 | 1200 | 1800 | 1300 | 2600 | 1100 | 600 | 600 |
| LiteGS-balance | 2000 | 1000 | 2000 | 1400 | 800 | 800 | 400 | 900 | 600 | 1200 | 600 | 1000 | 1000 |
| LiteGS-quality | 5987 | 3618 | 5728 | 4867 | 3770 | 1548 | 1190 | 1803 | 1252 | 2584 | 1085 | 3273 | 2326 |
| **Training Time (s)** | | | | | | | | | | | | | |
| TamingGS-budget | 353 | 306 | 565 | 261 | 349 | 348 | 401 | 469 | 371 | 224 | 252 | 272 | 227 |
| Mini-Splatting | 957 | 1020 | 1021 | 958 | 1027 | 1377 | 1642 | 1611 | 1381 | 747 | 765 | 1097 | 976 |
| Mini-Splatting v2 | 171 | 165 | 205 | 157 | 153 | 241 | 279 | 298 | 256 | 135 | 147 | 180 | 151 |
| LiteGS-turbo | 120 | 133 | 127 | 124 | 132 | 139 | 172 | 194 | 172 | 101 | 115 | 120 | 103 |
| 3DGS | 2070 | 1455 | 2156 | 1660 | 1487 | 1435 | 1416 | 1712 | 1246 | 1064 | 748 | 1677 | 1307 |
| TamingGS-big | 1263 | 862 | 1233 | 989 | 885 | 721 | 698 | 875 | 629 | 661 | 433 | 966 | 660 |
| 3DGS-MCMC | 3970 | 2973 | 3961 | 4018 | 3190 | 2622 | 2529 | 2766 | 2256 | 1906 | 1107 | 2371 | 2268 |
| SSS | 2834 | 3135 | 4427 | 1563 | 2809 | 5058 | 5409 | 6046 | 4368 | 3737 | 2398 | 1808 | 1737 |
| LiteGS-balance | 328 | 278 | 348 | 278 | 242 | 285 | 282 | 374 | 287 | 294 | 233 | 212 | 219 |
| LiteGS-quality | 548 | 584 | 693 | 555 | 507 | 372 | 467 | 527 | 389 | 479 | 317 | 347 | 314 |

Table 6: Quantitative results on Mip-NeRF 360 under the preprocessing setting of 3DGS-MCMC.

| Method | Mip-NeRF 360 | | |
| --- | --- | --- | --- |
| | SSIM↑/PSNR↑/LPIPS↓ | Train↓ | Num |
| 3DGS MCMC | 0.900 / 29.89 / 0.190 | 3126 | 3.09 |
| SSS | 0.893 / 29.90 / 0.145 | 4123 | 2.11 |
| LiteGS-balance | 0.888 / 29.90 / 0.155 | **314** | 1.15 |
| LiteGS-quality | **0.896 / 30.15 / 0.137** | 526 | 3.19 |

## B.4 3DGS-MCMC PREPROCESSING PIPELINE

Due to differences in pre-processing methods, the results of 3DGS-MCMC and SSS on the Mip-NeRF 360 dataset reported in Table 1 differ from those of their original publications. The 3DGS-MCMC preprocessing pipeline is as follows: (1) excluding the flowers and treehill scenes, and (2) training and testing on inputs downsampled by a factor of 4 using the PIL library from the original resolution (instead of using the dataset-provided downsampled images).

To ensure a fair comparison, we also provide comparative results in Table 6 using 3DGS-MCMC preprocessing. LiteGS-quality achieves the best overall quality, while LiteGS-balance demonstrates a strong speed-quality tradeoff with 10x faster training than 3DGS-MCMC.

