# OpenReview forum: "LiteGS: a high-performance framework to train 3dgs in subminutes via system and algorithm codesign"
_ICLR.cc/2026/Conference — ICLR 2026 Conference Desk Rejected Submission_

### Official Review · Reviewer_WH8U · 2025-10-20

**Soundness:** 3
**Presentation:** 2
**Contribution:** 2
**Rating:** 4
**Confidence:** 3

**Summary:**

This paper comprehensively analyze the 3DGS training system and detects related issues with the current paradigm. Based on these insights, it proposes a co-optimization for 3DGS systems, covering high-level algorithm design to low-level CUDA backbones, integrating GPU kernel design, memory layout, and algorithmic improvements. The resulting method considers fast training speed, compact storage size and high-quality reconstruction.

**Strengths:**

1. This paper is well-motivated and targets at a research gap where scare previous works exist. The 3DGS efficiency is of significance and requires comprehensive efforts for improvements.
2. This work integrates effective technical methods to improve the 3DGS training efficiency, including warp-based rasterization which accelerates tile-based rasterization, mixed precision training in 3DGS, cluster-cull-compaction with Morton code sorting, and opacity reset strategy considering variance as a metric.
3. The overall performance accelerates the 3DGS training process, makes the storage compact, while maintaining reasonable reconstruction quality.

**Weaknesses:**

1. The paper contributions and novelties are hard to judge. This paper includes multiple off-the-shelf techniques, such as scanline, integer warp reduce, and mixed precision, which are not first established while primarily combined in this work. These elements make the proposed method a good adaptation of the previous thoughts and techniques, while it is difficult to state their novelty compared to previous CUDA acceleration techniques.
2. Paper illustration. The CUDA operations in this paper are not intuitive to understand, and more visualizations are favorable for a better illustration.
3. Method effectiveness. This paper employs three model variants, turbo, balance and quality, with different model hyperparameters for different comparisons. Although these variants perform well in their own specific domain, this fails to verify that the proposed method achieves a superior efficiency-quality trade-off over baseline methods. The proposed method still relies on fine-grained adjustments to 'transfer' to different versions for better performance on partial metrics, leading to low confidence on a comprehensively effective method. When implementation, the users still need to state their requirements and select one variant compromising some other metrics. This leads to significant concerns on the effectiveness.

**Questions:**

1. The primary concern is on the method effectiveness. As illustrated in the weakness, the method is not a universally excellent solution to the efficiency-performance trade-off, and it requires multiple variants to show comparisons on different aspects. The authors are expected to provide justifications on this.
2. The authors are expected to provide more clarifications on its paper novelty, especially ones related to the GPU backbone.
3. It is recommended to provide more visual illustrations to aid paper readability. For example, the Supp Fig. 3 4 5 can be placed to the main paper part.

---

> ### Author Response · Authors · 2025-11-14
> **Addressing W3 & Q1: Efficiency–Performance Trade-off and Hyperparameter Fairness**
>
> Thank you very much for raising important questions regarding the efficiency–performance trade-off and the influence of inconsistent hyperparameters across baselines. We completely agree that evaluating parameter-efficiency requires clarity and careful control of confounding factors. We apologize if our initial presentation did not fully convey this. **In fact, one of the core intentions of LiteGS is to provide a stable and interpretable efficiency–performance trade-off across different parameter budgets.**
>
> # Why we introduce three LiteGS configurations (turbo / balance / quality)
> To avoid cherry-picking and to reflect the common usage patterns in recent 3DGS research, we adopt three representative parameter regimes:
>
> LiteGS-turbo: lightweight / constrained parameter budget, matching the design philosophy of Mini-Splatting v2 (18k iterations).
>
> LiteGS-balance: medium-scale budget, aiming to balance accuracy and speed; widely used in existing 3DGS works.
>
> LiteGS-quality: large-scale parameter budget comparable to original 3DGS and high-fidelity models (e.g., 3DGS-MCMC).
>
> These three regimes are not tuned to outperform specific baselines; they represent canonical model sizes widely used in the 3DGS community. Their purpose is to ensure that LiteGS is evaluated broadly and transparently across the full spectrum of practical budgets.
>
> # Why we align LiteGS to baseline hyperparameters rather than enforce a unified protocol
>
> The inconsistency in baseline hyperparameters arises from fundamental differences in method design:
>
> * **Different optimization goals** Lightweight methods intentionally reduce training iterations (e.g., Mini-Splatting v2: 30k→18k),
> while quality-oriented methods increase them (e.g., SSS with 40k iterations on MipNeRF360).
>
> * **Different densification mechanisms → different natural parameter counts** Original 3DGS grows parameters through threshold-based splitting, while later works (e.g., TamingGS) impose fixed parameter budgets. **Therefore, baselines do not share comparable default model sizes, as reflected in Table 5.**
>
> Forcing all baselines into a unified parameter budget would degrade their performance by overriding their intended design choices. To avoid unfairly weakening baselines, we therefore preserve their original hyperparameters and instead align LiteGS to each baseline’s parameter scale and iteration count. This ensures each baseline is evaluated under its best-known setting.
>
> # Why this evaluation protocol is fair and complete
>
> This protocol shows that even under each baseline’s own optimal configuration, LiteGS remains consistently faster and more accurate.
>
> * Table 5 provides each baseline’s fixed parameter size.
> * Figure 6 presents LiteGS performance continuously over a broad range of parameter budgets.
> * Reviewer can directly match any baseline’s parameter count to the corresponding LiteGS point.
>
> Thus, **Table 5 + Figure 6 already provide parameter-aligned, one-to-one comparisons**, without changing any baseline’s design or weakening it.

---

> > ### Author Response · Authors · 2025-11-14
> > **LiteGS provides a unified, robust, and efficient trade-off across all budgets**
> >
> > Your comment touches on an important conceptual point: In a fixed parameter regime, should one scale up a lightweight model, or scale down a high-quality model?
> >
> > Our findings suggest that LiteGS eliminates this dilemma. By combining robust densification, better parameter utilization, and system-level acceleration, LiteGS provides:
> >
> > * higher quality than lightweight models at the same or smaller parameter budget
> >
> > * higher quality with significantly less training time and fewer parameters than high-fidelity models.
> >
> > Thus, LiteGS establishes a consistent and unified efficiency–performance trade-off, independent of model size.

---

> > ### Comment · Reviewer_WH8U · 2025-11-24
> >
> > I appreciate the insightful response provided by the authors. These comments deals with my concern on the effectiveness evaluation. The authors clarify that the proposed method achieves universally good performances with various parameter settings, instead of cherry picking specific settings for a SOTA evaluation results.
> >
> > Based on these thinkings, I would like to raise my evaluation to a positive score.
> >
> > (However, as I have stated in the confidence part, I am not an expertise on the CUDA related designs and I am not fully confident or certain on these, which I hope that other reviewers or AC would give a better evaluation.)

---

> ### Author Response · Authors · 2025-11-14
> **Offering a unified evaluation protocol if requested**
>
> Although we believe that our current protocol—preserving each baseline’s original hyperparameters and aligning LiteGS accordingly—is the fairest way to avoid degrading baselines, we fully understand the reviewer’s concern regarding a completely unified experimental setting.
>
> To address this transparently, we would be very happy to provide an additional set of experiments under a fully standardized configuration (e.g., fixed parameter count, fixed iteration count, and shared learning rate schedule across all methods), if the reviewer considers such a unified protocol necessary. We did not include these results initially to avoid weakening baselines by overriding their published configurations, but we are fully willing to add them based on the reviewer’s preference.
>
> Please kindly let us know if you would like to see this unified evaluation; we will provide it promptly.

---

### Official Review · Reviewer_VD5m · 2025-10-29

**Soundness:** 3
**Presentation:** 3
**Contribution:** 3
**Rating:** 6
**Confidence:** 2

**Summary:**

This paper presents LiteGS, a high-performance framework that significantly accelerates the training of 3D Gaussian Splatting (3DGS) through a comprehensive, multi-layer co-optimization of system and algorithm design.
The authors identify key bottlenecks in 3DGS training — including inefficient gradient reduction, poor spatial data locality, and unstable densification — and address them via three major contributions:

Warp-based Rasterization: A “one warp per tile” raster paradigm that minimizes gradient reduction overhead by fusing intra-thread accumulation and a single warp-level reduction, complemented by a scanline-based data reuse scheme and strategic mixed-precision computation to mitigate register pressure.

Cluster-Cull-Compact Pipeline: A Morton-code-based spatial sorting and clustering strategy enabling efficient cluster-level culling and memory compaction, improving cache coherence and reducing warp divergence.

Opacity Gradient Variance Metric: A new densification criterion based on the variance of per-pixel opacity gradients, paired with an opacity decay mechanism that improves the robustness and precision of geometric refinement.

Empirical results show that LiteGS accelerates 3DGS training by up to 13.4× while maintaining or even improving rendering quality. For high-quality setups, it achieves up to 10.8× speedup over quality SOTA (3DGS-MCMC) with better PSNR. The framework is open-sourced and demonstrates superior performance in both lightweight and full-quality scenarios.

**Strengths:**

- The paper presents a systematic and multi-level optimization for 3D Gaussian Splatting training. While individual optimizations in GPU computation or densification have appeared in prior works, LiteGS’s holistic co-design across low-level GPU kernels, mid-level data management, and high-level algorithmic logic is novel. The introduction of a variance-based opacity gradient metric for densification is particularly innovative and conceptually elegant.
- The paper provides solid technical depth and engineering rigor. The proposed warp-based rasterization, mixed-precision accumulation, and Morton-code clustering are well justified and clearly described with mathematical precision. The methods are hardware-conscious yet algorithmically generalizable, demonstrating a rare balance between system-level and algorithmic contributions.
Experimental results are comprehensive, with ablation studies validating the necessity of each design choice and comparisons against multiple baselines (e.g., 3DGS, 3DGS-MCMC, Minisplatting v2).
- The paper is well structured and clearly written, with a logical flow from problem analysis to multi-layer solutions. The figures and appendices (as referenced) likely aid understanding of complex system details. The exposition effectively explains how low-level optimizations propagate benefits to higher-level algorithmic efficiency.

**Weaknesses:**

- While LiteGS is highly effective for 3DGS, it remains unclear whether the proposed techniques — especially the warp-based rasterization and Morton-based clustering — generalize well to other Gaussian-based rendering systems (e.g., Gaussian Surfels, Spec-Gaussians) or non-Gaussian volumetric methods. A brief discussion or experiment on this generalizability would strengthen the work’s impact.
- Although the paper mentions “comprehensive ablation studies,” it is unclear how the individual contributions interact (e.g., how much additional speedup is achieved when combining warp-based rasterization with clustering, beyond their individual effects). Providing a layer-wise ablation table could make the quantitative benefits more interpretable.
- The mixed-precision strategy may introduce numerical instability or minor visual artifacts, particularly in regions with fine opacity transitions. A quantitative error analysis (e.g., comparing FP32 vs. mixed-precision PSNR or SSIM) would make the claims of “comparable or superior quality” more convincing.
- The paper could benefit from a more detailed visualization of performance bottlenecks before optimization (e.g., profiling charts showing GPU utilization or cache miss rates). This would make the motivation more concrete for readers unfamiliar with GPU-level optimization.

**Questions:**

See Weaknesses

---

> ### Author Response · Authors · 2025-11-26
> **Response to Weakness 1**
>
> Thank you for your insightful comment regarding the generalizability of our proposed techniques. We clarify that unlike methods modifying the Gaussian splatting rendering formulation (such as MiniSplatting v2, SSS, etc.), LiteGS maintains mathematical equivalence with the original 3DGS.
>
> Regarding component integration:
>
> * Warp-based rasterization module is designed as a true drop-in replacement that completely substitutes the existing rasterizer without requiring any code modifications to the rendering pipeline. It maintains mathematical equivalence with the original renderer while delivering significant speed improvements.
>
> * Cluster-Cull-Compact pipeline, while involving training workflow modifications that prevent straightforward modular encapsulation, can be integrated into existing frameworks with reasonable engineering effort. The core spatial clustering concepts are well-established in computer graphics.
>
> Fundamental Parallelization Model:
>
> * warp-based rasterization represents a novel parallelization model rather than a technique limited to Gaussian primitives. Any rendering system employing transmittance-based primitives for approximate volume rendering could benefit from this parallelization strategy. The core insight—assigning an entire warp to process a coherent screen-space tile—addresses fundamental hardware utilization challenges that transcend specific primitive representations.

---

> ### Author Response · Authors · 2025-11-26
> **Response to Weakness 3**
>
> Thank you for raising the important question regarding potential numerical instability from our mixed-precision strategy. We conducted a comprehensive analysis to validate that half-precision rasterization introduces negligible error in 3DGS training. Our findings confirm that FP16 rasterization has no perceptible negative impact on training quality for the following fundamental reasons:
>
> 1. **8-bit RGB** : Ground truth images in dataset are 8-bit RGB. Rendering beyond this precision provides no perceptual benefit, as any error below 1/255 intensity unit is imperceptible.
>
> 2. **Algorithmic simplicity**: 3DGS rendering involves only alpha blending of Gaussian primitives without complex material interactions, BRDF evaluations, or multi-bounce lighting calculations that would amplify numerical errors.
>
> 3. **Existing precision thresholds**: The baseline implementation already employs aggressive precision truncation:
>     * fragments with opacity below 1/255 are discarded
>     * Alpha blending is truncated at transmittance values below 0.0001
>
> To quantitatively validate these observations, we performed a rigorous error analysis comparing FP32 and FP16 rasterization. Using FP32 rasterization gradients as ground truth, we measured RMSE (Root Mean Square Error) and MER (Mean Relative Error).
>
> |RMSE| MRE [1e-6,1e-4) | MRE [1e-4,1e-2) | MRE [1e-2, $ +\infty $)|
> |---|---|---|---|
> | 3.6974e-06 | 0.0665 | 0.0131 | 0.0035 |

---

> ### Author Response · Authors · 2025-11-27
> **Response to W2: Ablation**
>
> To thoroughly analyze the individual contributions of our system-level optimizations, we conducted a hierarchical ablation study as requested.
>
> Experimental Setup: To isolate the gains from the "Cluster & Cull & Compact" pipeline and the "Warp-based Rasterizer", we established a uniform baseline: the original 3DGS framework integrated with our proposed densification strategy. We evaluated four configurations using the litegs-quality setting:
> * baseline: Original 3DGS + Our Densification.
> * + Cluster&Cull&Compact: Baseline with our Morton-code based sorting and culling.
> * + Warp-based Raster: Baseline with our optimized rasterizer.
> * Full(LiteGS): Combining both optimizations.
>
> Results:
>
> |  | Mip-Nerf 360 (s) | Tanks&Temples (s) | Deep Blending (s) |
> | --- | ---| ---| ---|
> | Baseline | 1833 | 1042 | 1711 |
> | + Cluster&Cull&Compact | 1676 | 971 | 1557 |
> | + Warp-based Raster | 701 | 483 | 533 |
> | + Both(LiteGS) | 515| 398 | 330 |
>
> (Note on "Baseline" Training Time: You may notice that the "Baseline" training time reported here is slightly higher than the baseline in Table 2 of our main paper. This is because the baseline in this ablation utilizes our densification strategy, which involves more complex gradient variance calculations compared to the original 3DGS.)
>
> Analysis:
>
> * Correlation with Kernel Benchmarks: The end-to-end training results align with the kernel-level acceleration reported in Table 3. The Warp-based Rasterizer serves as the primary driver for total training time reduction.
>
> * Bottleneck Shift: On the baseline configuration, the relative speedup from the "Cluster & Cull" method is less pronounced. This is because the performance bottleneck is dominated by the heavy computation of the original rasterization process, which masks the benefits of efficient data management. However, once the Warp-based Rasterizer is applied, the rasterization bottleneck is alleviated, allowing the Cluster-Cull-Compact pipeline to contribute significant acceleration.
>
> * Synergistic Effect & Memory I/O: Comparing the absolute time reduction reveals a strong synergistic effect between the two methods. In Deep blending dataset, applying "Cluster & Cull" to the baseline reduces training time by **154s**, whereas applying it to the Warp-based Rasterizer results in a **203s** reduction. We attribute this amplified gain to a shift in system bottlenecks. The substantial acceleration provided by the Warp-based Rasterizer pushes the system towards the memory I/O bottleneck, where performance becomes significantly more sensitive to disordered memory access. This issue is particularly pronounced in outward-facing scenes (Deep Blending), where each view renders only a sparse subset of primitives scattered across a large, disordered global buffer. In this scenario, random memory access patterns severely hamper the efficiency of our rasterizer. "Cluster & Cull & Compact" addresses this by reordering primitives to maximize locality, thereby alleviating the I/O pressure and allowing the Warp-based Rasterizer to get its peak throughput.

---

### Official Review · Reviewer_48y1 · 2025-10-30

**Soundness:** 2
**Presentation:** 2
**Contribution:** 2
**Rating:** 2
**Confidence:** 3

**Summary:**

This paper proposes LiteGS, a novel 3D Gaussian Splatting (3DGS) framework designed to substantially reduce the computational overhead for 3DGS training. To this end, the authors introduce several efficiency-oriented techniques, including a warp-based rasterizer, cluster-cull-compact pipeline, and a novel densification metric. Experimental results demonstrate that the proposed method achieves high-quality reconstruction performance while significantly shortening training time.

**Strengths:**

- Warp-based rasterizer enables both fast forward and backward pass, leading to a significant reduction in training time.

- Morton curve-based clustering allows efficient cluster-level culling, resulting in computational benefits.

- This method achieves high rendering quality despite the minimal training costs.

**Weaknesses:**

- The overall manuscript is difficult to follow due to unclear writing and structure. Improving the notations, descriptions, and overall presentation would enhance readability.

- This method mainly focuses on hardware-related optimizations and therefore lacks sufficient technical novelty from a methodological perspective.

- Despite the high performance, the emphasis on hardware-level efficiency may not align well with the focus of this conference.

**Questions:**

- GPU memory usage is also an important factor of training efficiency. Could you clarify how the proposed method improves performance with respect to the memory footprint?

- The performance reported in Table 4 is not consistent with that shown in Table 1. Please clarify which variant corresponds to the scores in Table 4.

- Most of the techniques are related to training acceleration. Then, which component of this method contributes to the improvement in rendering quality, not the training efficiency, compared to existing 3DGS approaches?

- How to apply the proposed metric in densification? Could you explain more details on this part?

---

> ### Author Response · Authors · 2025-11-17
> **Rebuttal to R2**
>
> # W3
> We respectfully disagree with the concern that our work may fall outside the scope of the conference due to its focus on efficiency. Our submission explicitly selected the Primary Area: “applications to computer vision, audio, language, and other modalities,” which is fully aligned with the topic of 3D Gaussian Splatting.
>
> ICLR has consistently accepted works on 3D reconstruction, differentiable rendering, and 3DGS-related methods in recent years. Many of these works also include performance components, as efficiency is a central challenge in making these models practical for real-world applications.
>
> Our contribution fits squarely within this category. Although LiteGS introduces system-level innovations, its purpose and outcome are algorithmic improvements to 3DGS training—a widely used representation in computer vision and graphics. The method directly advances reconstruction quality, parameter efficiency, and training scalability, which are core scientific questions relevant to this area.
>
> # Q1
>
> Our method does not specifically target reducing memory footprint. However, LiteGS achieves significantly higher parameter efficiency, meaning that comparable or better reconstruction quality can be obtained with fewer Gaussians. This indirectly lowers memory usage.
>
> # Q2
>
> The results in Table 4 correspond to the LiteGS-balance configuration. The small numerical differences between Table 4 and Table 1  fall within the normal variation observed across multiple runs. No change of configuration or hyperparameters was involved.
>
> # Q3
>
> The improvement in rendering quality comes from our densification strategy, specifically the proposed opacity-gradient variance metric. This metric produces more accurate and stable splitting decisions, leading to significantly higher parameter efficiency. As a result, LiteGS achieves better reconstruction quality than existing 3DGS approaches under the same parameter budget.
>
> # Q4
>
> We apologize for not explicitly describing the densification pipeline in the main paper. Our densification process follows the standard 3DGS setting; we only change the score used to decide which Gaussians to split. Due to space limits we did not repeat the full procedure, so we briefly summarize it here and clarify how our metric is applied.
>
> During training, common 3DGS methods collect per-Gaussian signals such as gradient magnitude, depth, or reconstruction error to detect under-fit image regions. Because each Gaussian only influences a very local area, these regions cannot be corrected unless new Gaussians are added. The densification step therefore identifies under-represented areas and creates new Gaussians to improve reconstruction quality.
>
> A limitation of commonly used scores (gradient magnitude, accumulated error, etc.) is that they are only reliable when the underlying parameters are already close to convergence. However, 3DGS training periodically resets opacity to improve optimization, so we cannot assume that the parameters are near a converged state during most of training. Right after an opacity reset, these scores become poorly correlated with true under-fitting. To address this, we propose a simple and principled score: opacity-gradient variance across views.
>
> Densification is triggered at a fixed frequency. Every ten epochs, we accumulate opacity gradients for every Gaussian and compute their variance. We then sort all Gaussians by this variance and split the top-N candidates, where N is determined jointly by the parameter budget and the current number of Gaussians.

---

> ### Author Response · Authors · 2025-11-28
> **Additional Clarifications for Q3&Q4**
>
> Thank you again for taking the time to review our submission. We would like to provide an additional clarification that may address some of the concerns in your review.
>
> From your comments, we suspect that you might be particularly interested in a deeper examination of the densification mechanism and its quantitative impact. **In the rebuttal and in our response to Reviewer R1, we included two sets of extended experiments**:
>
> # Quantitative Precision Analysis:
>
> We designed a metric to evaluate whether the primitives selected by our method are indeed the ones that contribute most to the reconstruction error. We define the Densification Precision as $P = \frac{|A \cap B|}{|A|}$
>
> * set A: The Top-10% primitives selected by the densification strategy (ours or baseline) at early training stages.
> * set B: The set of primitives that exhibit the highest accumulated error (gradients/loss) after the model has fully converged (e.g., at epoch 20). This serves as a "ground truth" proxy for regions that are inherently difficult to fit. This serves as a "ground truth" proxy for regions that are inherently difficult to fit [1, 2].
>
> As summarized in the table below, we compare the Densification Precision at the beginning of training. It is crucial to emphasize that the challenge of densification precision exists not only during the beginning of training but also following opacity resets, which occur periodically in standard 3DGS training. Our method demonstrates consistently superior precision in these critical phases (e.g., reaching 64.2% at epoch 8 compared to only 13.7% for TamingGS).
>
> | epoch | 2 | 4 | 8 |
> | ---- |----|----|---|
> | Ours | 49.4% | 56.1% | 64.2% |
> | TamingGS | 8.6% | 11.1% | 13.7% |
>
> We acknowledge that the use of different metrics for the selection candidates (Set A) and the ground truth (Set B) might raise questions. However, we emphasize that densification metrics in baselines like TamingGS are primarily heuristic; they lack theoretical interpretability and fail to effectively label under-fitted regions after the model has converged. Therefore, it is necessary to define Set B using the theoretically grounded metrics from GOF [1] and Revisiting 3DGS [2], rather than the heuristics of the baselines. To address any remaining uncertainties regarding the choice of these proxy metrics and to conclusively demonstrate the robustness of our approach, we provide an additional end-to-end stress test below.
>
> [1] Rota Bulò S, Porzi L, Kontschieder P. Revising densification in gaussian splatting[C]//European Conference on Computer Vision. Cham: Springer Nature Switzerland, 2024: 347-362.
>
> [2] Yu Z, Sattler T, Geiger A. Gaussian opacity fields: Efficient adaptive surface reconstruction in unbounded scenes[J]. ACM Transactions on Graphics (ToG), 2024, 43(6): 1-13.
>
> # Aggressive Stress Test & New Efficiency Frontier
>
> To demonstrate the stability and accuracy of our densification strategy under constrained conditions, we performed an experiment with an aggressive setup based on the litegs-turbo configuration. We halved the total training iterations while forcing the model to target 2x the number of parameters. This requires the algorithm to rapidly and accurately identify where to add a massive number of parameters in a very short time.
>
>  **Results** :
>
> As shown in the table below, this experiment yields two critical observations regarding stability and efficiency:
>
> * Stability: TamingGS fails to converge entirely under this aggressive parameter growth schedule, confirming the fragility of heuristic strategies when pushed to the limit.
>
> * Breakthrough: In stark contrast, LiteGS remains **stable** and delivers **a remarkable trade-off**. On the Mip-NeRF 360 dataset, compared to litegs-turbo, the results of Stress Test incurs only a marginal quality drop (0.2 dB in PSNR) but slashes the training time by approximately 50%.
>
> |method| prmitives(k) | takes time(s) | SSIM | PSNR | LPIPS |
> |--- |---|---|---|---|---|
> | LiteGS | 1.171 |80.98 |	0.817 |	27.57 |	0.221 |
> | TamingGS | 1.171 | 161.13 | 0.253 | 11.23 | 0.807 |
>
>
> This result does more than prove stability; it demonstrates that LiteGS establishes a new efficiency frontier, making high-quality 3D reconstruction possible in mere seconds—a capability unattainable by previous baselines.

---

### Official Review · Reviewer_LPDX · 2025-10-31

**Soundness:** 4
**Presentation:** 3
**Contribution:** 4
**Rating:** 6
**Confidence:** 2

**Summary:**

This paper tackles the critical bottleneck of slow training times in 3D Gaussian Splatting (3DGS), which currently hinders its broader application. The authors propose LiteGS, a high-performance framework that systematically optimizes the entire 3DGS training pipeline through a novel "system and algorithm codesign" approach.

**Strengths:**

1. The performance gain is very impressive.
2. Deep and Novel Technical Contributions: Each of the three core contributions is substantial and novel in its own right.
- The Warp-based Rasterizer is an excellent piece of systems engineering, demonstrating a hardware-aware design that cleverly uses scanline algorithms and mixed-precision (including the novel use of integer warp-reduce for float accumulation) to solve the specific bottlenecks of 3DGS backpropagation.
- The Cluster-Cull-Compact pipeline is a smart adaptation of a known (but from a different domain) technique to the dynamic training of 3DGS, effectively solving the cache-locality and warp-divergence problems that worsen as the model grows.

**Weaknesses:**

The "opacity gradient variance" metric is a key contribution to improving PSNR, as demonstrated through extensive experiments in the paper. However, it would be helpful if the authors could provide additional visualizations of the Gaussian distributions (e.g., gradient histograms, variance heatmaps) to better illustrate the concept and assist the reviewer in understanding the method.

Nevertheless, I am not an expert in 3D-GS rendering pipelines, so I would defer to other reviewers’ opinions on this aspect.

**Questions:**

I have no more questions.

---

> ### Author Response · Authors · 2025-11-25
> **Effectiveness of Densification Metric**
>
> We thank the reviewer for the insightful question. To rigorously validate that our Opacity Gradient Variance metric effectively identifies under-fitted regions (and is not merely a heuristic), we conducted two additional experiments: a quantitative precision analysis and an aggressive end-to-end stress test.
>
> # Quantitative Precision Analysis
>
> We designed a metric to evaluate whether the primitives selected by our method are indeed the ones that contribute most to the reconstruction error. We define the Densification Precision as $P = \frac{|A \cap B|}{|A|}$
>
> * set A: The Top-10% primitives selected by the densification strategy (ours or baseline) at early training stages.
> * set B: The set of primitives that exhibit the highest accumulated error (gradients/loss) after the model has fully converged (e.g., at epoch 20). This serves as a "ground truth" proxy for regions that are inherently difficult to fit. This serves as a "ground truth" proxy for regions that are inherently difficult to fit [1, 2].
>
> As summarized in the table below, we compare the Densification Precision at the beginning of training. It is crucial to emphasize that the challenge of densification precision exists not only during the beginning of training but also following opacity resets, which occur periodically in standard 3DGS training. Our method demonstrates consistently superior precision in these critical phases (e.g., reaching 64.2% at epoch 8 compared to only 13.7% for TamingGS).
>
> | epoch | 2 | 4 | 8 |
> | ---- |----|----|---|
> | Ours | 49.4% | 56.1% | 64.2% |
> | TamingGS | 8.6% | 11.1% | 13.7% |
>
> We acknowledge that the use of different metrics for the selection candidates (Set A) and the ground truth (Set B) might raise questions. However, we emphasize that densification metrics in baselines like TamingGS are primarily heuristic; they lack theoretical interpretability and fail to effectively label under-fitted regions after the model has converged. Therefore, it is necessary to define Set B using the theoretically grounded metrics from GOF [1] and Revisiting 3DGS [2], rather than the heuristics of the baselines. To address any remaining uncertainties regarding the choice of these proxy metrics and to conclusively demonstrate the robustness of our approach, we provide an additional end-to-end stress test below.
>
> [1] Rota Bulò S, Porzi L, Kontschieder P. Revising densification in gaussian splatting[C]//European Conference on Computer Vision. Cham: Springer Nature Switzerland, 2024: 347-362.
>
> [2] Yu Z, Sattler T, Geiger A. Gaussian opacity fields: Efficient adaptive surface reconstruction in unbounded scenes[J]. ACM Transactions on Graphics (ToG), 2024, 43(6): 1-13.
>
> # Aggressive Stress Test & New Efficiency Frontier
>
> To demonstrate the stability and accuracy of our densification strategy under constrained conditions, we performed an experiment with an aggressive setup based on the litegs-turbo configuration. We halved the total training iterations while forcing the model to target 2x the number of parameters. This requires the algorithm to rapidly and accurately identify where to add a massive number of parameters in a very short time.
>
>  **Results** :
>
> As shown in the table below, this experiment yields two critical observations regarding stability and efficiency:
>
> * Stability: TamingGS fails to converge entirely under this aggressive parameter growth schedule, confirming the fragility of heuristic strategies when pushed to the limit.
>
> * Breakthrough: In stark contrast, LiteGS remains **stable** and delivers **a remarkable trade-off**. On the Mip-NeRF 360 dataset, compared to litegs-turbo, the results of Stress Test incurs only a marginal quality drop (0.2 dB in PSNR) but slashes the training time by approximately 50%.
>
> |method| prmitives(k) | takes time(s) | SSIM | PSNR | LPIPS |
> |--- |---|---|---|---|---|
> | LiteGS | 1.171 |80.98 |	0.817 |	27.57 |	0.221 |
> | TamingGS | 1.171 | 161.13 | 0.253 | 11.23 | 0.807 |
>
>
> This result does more than prove stability; it demonstrates that LiteGS establishes a new efficiency frontier, making high-quality 3D reconstruction possible in mere seconds—a capability unattainable by previous baselines.

---

> ### Author Response · Authors · 2025-11-25
> **The detail results of our stress test**
>
> |scene|primitives|takes|SSIM_train|PSNR_train|LPIPS_train|SSIM_test|PSNR_test|LPIPS_test|
> |----|-----------|------|---------|---------|---------|---------|---------|---------|
> |bicycle|1360000|68.90290451049805|0.7696665|23.9320354|0.2501733|0.7585854|25.1978111|0.2335179|
> |flowers|1220000|78.57123351097107|0.7225419|23.1023922|0.2838508|0.6053527|21.7142277|0.3396027|
> |garden|1460000|78.55829739570618|0.8845506|28.7087154|0.1193212|0.8556744|27.3950653|0.1323125|
> |stump|1340000|71.96337366104126|0.8554742|28.5616417|0.2056226|0.7926696|27.2177505|0.2127895|
> |treehill|1160000|74.03676867485046|0.7261154|22.7449684|0.3029707|0.6390569|22.8510647|0.3379321|
> |room|800000|70.2774748802185|0.9351271|33.6474419|0.2011591|0.9221826|31.5905571|0.2149457|
> |counter|800000|89.81700301170349|0.9220967|29.9026985|0.1850652|0.9086227|28.8188915|0.2000091|
> |kitchen|1200000|105.24096488952637|0.9389035|32.4555969|0.1205899|0.9264742|31.4408417|0.1294753|
> |bonsai|1200000|91.47094392776489|0.9501833|32.8006744|0.1922134|0.9444824|31.9543724|0.1964863|
> |truck|680000|61.99935531616211|0.8910525|26.4909763|0.1484901|0.8747544|25.3993835|0.1531686|
> |train|720000|69.77397298812866|0.8102806|23.553236|0.2373699|0.7795711|21.1193752|0.2538348|
> |drjohnson|1600000|66.38695478439331|0.9353783|34.0604973|0.2259799|0.9080961|29.6174507|0.2518271|
> |playroom|980000|56.96188426017761|0.9448828|35.7014847|0.2159965|0.9150408|30.9205952|0.2443539|

---

### Author Response · Authors · 2025-11-17
**Novelty Rebuttal (for R2W2, R4W1，R4Q2)**

We fully agree that simply combining known GPU techniques such as scanline, warp reductions, or mixed precision would not constitute meaningful novelty. LiteGS is not a collection of such isolated components. Its main contribution is a unified algorithm–system co-design framework that addresses a set of coupled bottlenecks in 3DGS training. These bottlenecks were previously treated as unrelated, yet they directly constrain each other. Below, we clarify why LiteGS is technically non-trivial and cannot be reduced to reusing existing optimizations.

# LiteGS is the first framework that constructs a closed-loop, cross-layer bottleneck diagnosis for 3DGS training

Our analysis shows that the main limitations in 3DGS training do not stem from any single component. They arise from the interaction among densification, spatial/memory locality, and rasterization parallelism. Prior work improved these steps independently. LiteGS is the first to describe and optimize them jointly:

1. **Densification progressively breaks spatial and memory locality.** As densification begining, new primitives are appended to memory, breaking memory locality. LiteGS restores locality through online cluster-level reorganization. This enables cluster-level culling and avoids substantial redundant computation. Earlier methods do not maintain locality after densification and therefore incur unnecessary cost.

2. **Warp-based rasterization facing hardware I/O limits without locality restoration.** While warp-based raster increases throughput, it also exposes memory bandwidth limits. If spatial locality is degraded, the rasterizer suffers from cache misses, and warp stalls. With our locality restoration, warp-based rasterization can actually reach its intended performance.

3. **Warp-based rasterization enables a new class of densification strategies —— gradient statistics instead of sheuristic or sampled scoring.** Our densification metric computes opacity-gradient variance across all views. This requires fast, dense multi-pixel accumulation. Traditional 3DGS pipelines cannot do this efficiently; they rely on heuristics or sample only 10–20 views because pixel-level accumulation is too slow. The limitation is systemic, not algorithmic. With warp-based rasterization, dense statistics become feasible, making our densification formulation practical.

In summary, LiteGS is not a collection of isolated optimizations, but the first framework to jointly address this densify → locality → rasterization → densify closed-loop bottleneck, which prior works neither recognized nor attempted to unify.

# Warp-per-tile rasterization is a new parallelization model, not a low-level trick

The novelty is not the use of warp reductions or scanline in isolation. It is the mapping that assigns an entire warp to a coherent screen-space region. Recent works explore other mappings—per-splat (TamingGS), tensor-core groups (TC-GS), or block-per-tile (DistWard)—but none use warp-per-tile mapping.

If warp-per-tile rasterization were a straightforward combination of scanline, mixed-precision, and warp reduction, prior 3DGS works—which already use these components—would have converged to the same design. They did not. The difficulty lies in making this mapping viable: controlling register footprint, aligning memory behavior, and inter-tile accumulation. LiteGS resolves these difficulty and demonstrates our warp-based raster.

# A densification metric that corrects a long-standing issue

Our densification metric is a new algorithmic contribution, not an engineering trick. LiteGS introduces a intuitive and effective metric for the densification. We identified a a long-standing issue in 3DGS densification. The periodic opacity reset distorts gradient magnitude and leads to incorrect densification decisions. To correct this, we propose a simple and elegant metric—opacity gradient variance.
Importantly, computing this variance efficiently requires fast, dense statistics, which previous rasterization methods could not provide. Our warp-based rasterization makes this feasible, enabling gradient aggregation at low cost.

# Mathematical compatibility with the 3DGS ecosystem

LiteGS intentionally preserves the original 3DGS formulation. This allows LiteGS to serve as drop-in infrastructure for the entire ecosystem (AnySplat, SSS, ImageGaussian, and others). This design choice ensures broad compatibility and fast adoption. It should not be interpreted as a lack of novelty; rather, it reflects our goal to improve the system foundation without altering the mathematical model.

---

### Author Response · Authors · 2025-11-25
**Significant Breakthrough: training 3DGS in about 50 seconds**

We are delighted to report a major breakthrough achieved during our response to Reviewer 1's questions, which led us to perform a new set of aggressive experiments focusing on our densification strategy and system stability.

We extend our sincere gratitude to Reviewer 1 for their insightful query, which inadvertently helped us break through an inherent cognitive bias. By strictly following the training parameter settings of existing baselines, **we had unknowingly capped the full potential of LiteGS**.

Our core contribution—combining a superior, highly stable densification strategy with fast operators—fundamentally changes the traditional trade-off between the number of parameters and the necessary number of training iterations. To fully unlock our framework's capacity, we recognized the need to abandon conservative settings and choose a radically aggressive training configuration.

# The Extreme Efficiency Configuration

As demonstrated in our experiments for Reviewer 1, our densification strategy achieves significantly superior precision compared to existing methods. Leveraging this core strength, we adopted an aggressive densification schedule to accelerate the Gaussian splitting process, enabling us to quickly conclude this phase and drastically reduce the overall number of training iterations.

Our initial tests indicated that using **half the iteration count of the litegs-turbo configuration** could already match the quality achieved by the original 3DGS baseline. To compensate for the minor loss of primitive efficiency caused by this rapid densification, we implemented a "space for time" strategy: we **set the final Gaussian count to double that of the litegs-turbo configuration.** This aggressive Configuration resulted in a total training time of approximately 50 seconds.

# Results: training 3DGS in about 50 seconds!

This aggressive configuration resulted in a new standard for 3DGS training speed, achieving high-quality reconstruction in mere seconds:

|scene|primitives|takes time(RTX 3090)|takes time(RTX 4090)|SSIM_train|PSNR_train|LPIPS_train|SSIM_test|PSNR_test|LPIPS_test|
|---|---|---|---|---|---|---|---|---|---|
|bicycle|1360000|68.90290451049805|**41.72691751**|0.7696665|23.9320354|0.2501733|0.7585854|25.1978111|0.2335179|
|flowers|1220000|78.57123351097107|**47.9060986**|0.7225419|23.1023922|0.2838508|0.6053527|21.7142277|0.3396027|
|garden|1460000|78.55829739570618|**46.76381636**|0.8845506|28.7087154|0.1193212|0.8556744|27.3950653|0.1323125|
|stump|1340000|71.96337366104126|**44.23456001**|0.8554742|28.5616417|0.2056226|0.7926696|27.2177505|0.2127895|
|treehill|1160000|74.03676867485046|**46.34997225**|0.7261154|22.7449684|0.3029707|0.6390569|22.8510647|0.3379321|
|room|800000|70.27747488|**41.05782461**|0.9351271|33.6474419|0.2011591|0.9221826|31.5905571|0.2149457|
|counter|800000|89.81700301170349|**51.63395095**|0.9220967|29.9026985|0.1850652|0.9086227|28.8188915|0.2000091|
|kitchen|1200000|105.24096488952637|**63.1258564**|0.9389035|32.4555969|0.1205899|0.9264742|31.4408417|0.1294753|
|bonsai|1200000|91.47094392776489|**56.54596996**|0.9501833|32.8006744|0.1922134|0.9444824|31.9543724|0.1964863|
|truck|680000|61.99935531616211|**40.03852582**|0.8910525|26.4909763|0.1484901|0.8747544|25.3993835|0.1531686|
|train|720000|69.77397298812866|**46.26339579**|0.8102806|23.553236|0.2373699|0.7795711|21.1193752|0.2538348|
|drjohnson|1600000|66.38695478439331|**39.18569803**|0.9353783|34.0604973|0.2259799|0.9080961|29.6174507|0.2518271|
|playroom|980000|56.96188426017761|**32.52357793**|0.9448828|35.7014847|0.2159965|0.9150408|30.9205952|0.2443539|

Under this aggressive configuration, LiteGS maintains stability and achieves this dramatic speedup with only a marginal loss in PSNR (approximately $0.2$ to $0.3$ dB).

**For full verifiability and reproducibility, the corresponding configuration files and implementation details have been included in the updated Supplementary Material.**

---

### Author Response · Authors · 2025-12-03
**Rebuttal Summary for AC**

We would like to provide a brief summary of the rebuttal process.

# Reviewer 1

**R1 asked for quantitative evidence.** We provided the requested quantitative experiment, which we believe sufficiently addresses R1’s question. **Inspired by the quantitative study requested by R1, we further explored more aggressive training configurations for LiteGS.** This led to a meaningful discovery: we were able to **reduce the training time to 50 seconds**, achieving several-fold speedup over the current SOTA, while attaining accuracy higher than original 3DGS. This result, although obtained during rebuttal, reinforces the strength and potential impact of our method.

# Reviewer 2

We responded thoroughly to all questions raised by R2. We believe **R2’s concerns stem from misunderstandings of our method**. We provided detailed clarifications at the earliest opportunity, but **R2 did not respond further**.

# Reviewer 3

**R3 requested specific comparison experiments**, which we provided in the rebuttal. These experiments directly address R3’s concerns.

# Reviewer 4

**R4 initially questioned the fairness**, noting that multiple parameter settings were used for LiteGS. We clarified that LiteGS itself is designed to run with different training configurations across scenes; enforcing a single unified configuration on all baselines would unfairly degrade other methods. **Our one-to-one comparisons are clearly provided in Table 5 and Figure 6**, showing that our evaluations are indeed fair and consistent. After understanding this, **R4 revised their score from 4 to 6.**

---

### Note · Program_Chairs · 2026-01-17
**Submission Desk Rejected by Program Chairs**

The following references in this submission do not refer to real documents and/or have major errors in bibliographic information:

 Miroslav Hapala, Jiˇr´ı Bittner, and Vlastimil Havran. Towards practical real-time bvh building. In
Eurographics Symposium on Rendering, pp. 1–8, 2011.